# Distributions as Actions: A Unified Framework for Diverse Action Spaces

**Jiamin He**
Department of Computing Science
University of Alberta
Alberta Machine Intelligence Institute (Amii)
jiamin12@ualberta.ca

**A. Rupam Mahmood & Martha White**
Department of Computing Science
University of Alberta
Canada CIFAR AI Chair, Amii
{armahmood,whitem}@ualberta.ca

## ABSTRACT

We introduce a novel reinforcement learning (RL) framework that treats parameterized action distributions as actions, redefining the boundary between agent and environment. This reparameterization makes the new action space continuous, regardless of the original action type (discrete, continuous, hybrid, etc.). Under this new parameterization, we develop a generalized deterministic policy gradient estimator, *Distributions-as-Actions Policy Gradient* (DA-PG), which has *lower variance* than the gradient in the original action space. Although learning the critic over distribution parameters poses new challenges, we introduce *Interpolated Critic Learning* (ICL), a simple yet effective strategy to enhance learning, supported by insights from bandit settings. Building on TD3, a strong baseline for continuous control, we propose a practical actor-critic algorithm, *Distributions-as-Actions Actor-Critic* (DA-AC). Empirically, DA-AC achieves competitive performance in various settings across discrete, continuous, and hybrid control.[1]

## 1 INTRODUCTION

Reinforcement learning (RL) algorithms are commonly categorized into value-based and policy-based methods. Value-based methods, such as Q-learning (Watkins & Dayan, 1992) and its variants like DQN (Mnih et al., 2015), are particularly effective in discrete action spaces due to the feasibility of enumerating and comparing action values. In contrast, policy-based methods are typically used for continuous actions, though they can be used for both discrete and continuous action spaces (Williams, 1992; Sutton et al., 1999).

Policy-based methods are typically built around the policy gradient theorem (Sutton et al., 1999), with different approaches to estimate this gradient. The likelihood-ratio (LR) estimator can be applied to arbitrary action distributions, including discrete ones. In continuous action spaces, one can alternatively compute gradients via the action-value function (the critic), leveraging its differentiability with respect to actions. This idea underlies the deterministic policy gradient (DPG) algorithms (Silver et al., 2014) and the use of the reparameterization (RP) trick for stochastic policies (Heess et al., 2015; Haarnoja et al., 2018). These approaches can produce lower-variance gradient estimates by backpropagating through the critic and the policy (Xu et al., 2019).

Despite the flexibility of policy gradient methods, current algorithms remain tightly coupled to the structure of the action space. In particular, different estimators and architectures are often required for discrete versus continuous actions, making it difficult to design unified algorithms that generalize across domains. Although the LR estimator is always applicable, it often requires different critic architectures for different action spaces (Haarnoja et al., 2018; Christodoulou, 2019) and carefully designed baselines to manage high variance (Greensmith et al., 2004).

In this paper, we introduce the *distributions-as-actions framework*, an alternative to the classical RL formulation that treats the parameters of parameterized distributions as actions. For a Gaussian policy, for example, the distribution parameters are the mean and variance, while for a softmax policy they are the probability values. The RL agent outputs these distribution parameters to the

---

[1]Code is available at https://github.com/hejm37/da-ac.

environment, and the action sampling becomes part of the stochastic transition. Importantly, distribution parameters are typically continuous, even when the underlying actions are discrete, hybrid, or structured. By shifting the agent-environment boundary in this way, we can develop a single continuous-action algorithm for a diverse class of action spaces.[2]

To develop algorithms under the new framework, we first propose the *Distributions-as-Actions Policy Gradient* (DA-PG) estimator, and prove it has lower variance than the corresponding updates in the original action space. This reduction in variance can increase the bias, because the critic can be harder to learn. We develop an augmentation approach, called *Interpolated Critic Learning* (ICL), to improve this critic learning. We then introduce a deep RL algorithm based on TD3 (Fujimoto et al., 2018), called *Distributions-as-Actions Actor-Critic* (DA-AC), that incorporates the DA-PG estimator and ICL. We empirically evaluate DA-AC to assess the viability of this new framework and its ability to use a single algorithm across diverse action spaces. DA-AC achieves competitive and sometimes better performance compared to baselines in a variety of settings across continuous, discrete, and hybrid control. We also provide targeted experiments to understand the bias-variance trade-off of different gradient estimators, and show the utility of ICL for improving critic learning.

## 2 PROBLEM FORMULATION

We consider a Markov decision process (MDP) $\langle \mathcal{S}, \mathcal{A}, p, d_0, r, \gamma \rangle$, where $\mathcal{S}$ is the state space, $\mathcal{A}$ is the action space, $p : \mathcal{S} \times \mathcal{A} \to \Delta(\mathcal{S})$ is the transition function, $d_0 \in \Delta(\mathcal{S})$ is the initial state distribution, $r : \mathcal{S} \times \mathcal{A} \to \Delta(\mathbb{R})$ is the reward function, and $\gamma$ is the discount factor. Here, $\Delta(\mathcal{X})$ denotes the set of distributions over a set $\mathcal{X}$. We use $\pi(a|s)$ to represent the probability of taking action $a \in \mathcal{A}$ under state $s \in \mathcal{S}$ for policy $\pi$. The goal of the agent is to find a policy $\pi$ under which the below objective is maximized:

$$J(\pi) \doteq \sum_{t=0}^{\infty} \mathbb{E}_{S_0 \sim d_0, A_t \sim \pi(\cdot|S_t), S_{t+1} \sim p(\cdot|S_t, A_t)} \left[ \gamma^t R_{t+1} \right] = \sum_{t=0}^{\infty} \mathbb{E}_\pi \left[ \gamma^t R_{t+1} \right], \tag{1}$$

where the second formula uses simplified notation that we follow in the rest of the paper. The *(state-)value function* and *action-value function* of the policy are defined as follows:

$$v_\pi(s) \doteq \sum_{t=0}^{\infty} \mathbb{E}_\pi \left[ \gamma^t R_{t+1} | S_0 = s \right], \quad q_\pi(s, a) \doteq \mathbb{E}_\pi \left[ R_1 + \gamma v_\pi(S_1) | S_0 = s, A_0 = a \right]. \tag{2}$$

In this paper, we consider actor-critic methods that learns a parameterized policy, denoted by $\pi_{\boldsymbol{\theta}}$, and a parameterized action-value function, denoted by $Q_{\mathbf{w}}$. Given a transition $\langle S_t, A_t, R_{t+1}, S_{t+1} \rangle$, $Q_{\mathbf{w}}$ is usually learned using temporal-difference (TD) learning:

$$\mathbf{w} \leftarrow \mathbf{w} + \alpha \left( R_{t+1} + \gamma Q_{\mathbf{w}}(S_{t+1}, A_{t+1}) - Q_{\mathbf{w}}(S_t, A_t) \right) \nabla Q_{\mathbf{w}}(S_t, A_t), \tag{3}$$

where $\alpha$ is the step size, and $A_{t+1}$ is sampled from the current policy: $A_{t+1} \sim \pi_{\boldsymbol{\theta}}(\cdot|S_{t+1})$.

The policy is typically optimized using a surrogate of Equation (1):

$$\hat{J}(\pi_{\boldsymbol{\theta}}) = \mathbb{E}_{S_t \sim d, A_t \sim \pi_{\boldsymbol{\theta}}(\cdot|S_t)} \left[ Q_{\mathbf{w}}(S_t, A_t) \right], \tag{4}$$

where $d \in \Delta(\mathcal{S})$ is some distribution over states. Below we outline three typical estimators for the gradient of this objective.

**The likelihood-ratio (LR) policy gradient estimator** (Sutton et al., 1999) uses $\hat{\nabla}_{\boldsymbol{\theta}} \hat{J}(\pi_{\boldsymbol{\theta}}; S_t, A) = \nabla_{\boldsymbol{\theta}} \log \pi_{\boldsymbol{\theta}}(A|S_t) Q_{\mathbf{w}}(S_t, A)$, where $A \sim \pi_{\boldsymbol{\theta}}(\cdot|S_t)$. Since the LR estimator suffers from high variance, it is often used with the value function as a baseline:

$$\hat{\nabla}_{\boldsymbol{\theta}}^{\text{LR}} \hat{J}(\pi_{\boldsymbol{\theta}}; S_t, A) = \nabla_{\boldsymbol{\theta}} \log \pi_{\boldsymbol{\theta}}(A|S_t) \left( Q_{\mathbf{w}}(S_t, A) - V(S_t) \right), \tag{5}$$

where $V(S_t)$ could either be parameterized and learned or be calculated analytically from $Q_{\mathbf{w}}$ when the action space is discrete and low dimensional.

**The deterministic policy gradient (DPG) estimator** (Silver et al., 2014) is used when the action space is continuous and the policy is deterministic ($\pi_{\boldsymbol{\theta}} : \mathcal{S} \to \mathcal{A}$), and uses the gradient of $Q_{\mathbf{w}}$ with respect to the action:

$$\hat{\nabla}_{\boldsymbol{\theta}}^{\text{DPG}} \hat{J}(\pi_{\boldsymbol{\theta}}; S_t) = \nabla_{\boldsymbol{\theta}} \pi_{\boldsymbol{\theta}}(S_t)^\top \nabla_A Q_{\mathbf{w}}(S_t, A)|_{A=\pi_{\boldsymbol{\theta}}(S_t)}. \tag{6}$$

---

[2]Concurrently, Todorov (2025b) studied a related formulation that considers a similar shift in the agent-environment boundary. Their work focuses on establishing a theoretical equivalence between stochastic and deterministic policy gradients in continuous action settings. In contrast, we emphasize algorithm design and empirical evaluation across diverse action types. A more detailed discussion is provided in Appendix A.

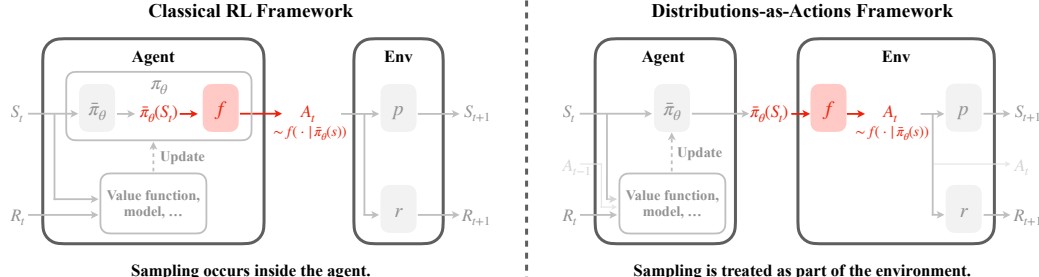

Figure 1: **Comparison between the classical reinforcement learning (RL) framework and the proposed distributions-as-actions framework.** In the classical RL setting (col 1), the agent's policy $\pi_{\theta}$ consists of $\bar{\pi}_{\theta}$, which produces the distribution parameters, and a sampling function $f$ that returns an action given these parameters. In the *distributions-as-actions framework* (col 2), the sampling function $f$ is considered part of the environment, and the agent outputs the distribution parameters $\bar{\pi}_{\theta}(S_t)$ as its action. The sampled action $A_t$ may optionally be observable to the agent, though it is not required for the core formulation. This shift redefines the interface between agent and environment, potentially simplifying learning and enabling new algorithmic perspectives.

**The reparameterization (RP) policy gradient estimator** (Heess et al., 2015) can be used if the policy can be reparameterized (i.e., $A = g_{\theta}(\epsilon; S_t)$, $\epsilon \sim p(\cdot)$, where $p(\cdot)$ is a prior distribution):

$$\hat{\nabla}_{\boldsymbol{\theta}}^{\mathrm{RP}} \hat{J}(\pi_{\boldsymbol{\theta}}; S_t, \epsilon) = \nabla_{\boldsymbol{\theta}} g_{\boldsymbol{\theta}}(\epsilon; S_t)^{\top} \nabla_A Q_{\mathbf{w}}(S_t, A)|_{A=g_{\boldsymbol{\theta}}(\epsilon; S_t)}. \tag{7}$$

## 3  DISTRIBUTIONS-AS-ACTIONS FRAMEWORK

The action space is typically defined by the environment designer based on domain-specific knowledge. Depending on the problem, it may be more natural to model the action space as discrete, continuous, or a hybrid of both. In all cases, the agent's policy at a given state $s$ can often be interpreted as first producing distribution parameters $\bar{\pi}_{\theta}(s)$, followed by sampling an action $A \sim f(\cdot|\bar{\pi}_{\theta}(s))$ from the resulting distribution. With a slight abuse of notation, we denote $\bar{\pi}_{\theta} : \mathcal{S} \to \mathcal{U}$ as the part of the policy $\pi_{\theta}$ that maps states to distribution parameters, and $f(\cdot|u)$ as the distribution over actions parameterized by $u \in \mathcal{U}$. Note that $\mathcal{U}$ is usually continuous.

In the classical RL framework, both $\bar{\pi}_{\theta}$ and $f$ are considered part of the agent, as in the left of Figure 1. In this work, we introduce the *distributions-as-actions framework*: the agent outputs distribution parameters $\bar{\pi}_{\theta}(s)$ as its action, while the sampling process $A \sim f(\cdot|\bar{\pi}_{\theta}(s))$ is treated as part of the environment, depicted on the right in Figure 1. This reformulation leads to a new MDP in which the action space is the distribution parameter space $\mathcal{U}$. The transition and reward functions in this MDP become

$$\bar{p}(s'|s, u) \doteq \mathbb{E}_{A \sim f(\cdot|u)}\big[p(s'|s, A)\big], \quad \bar{r}(s, u) \doteq \mathbb{E}_{A \sim f(\cdot|u)}\big[r(s, A)\big]. \tag{8}$$

This gives rise to the *distributions-as-actions MDP* (DA-MDP) $\langle \mathcal{S}, \mathcal{U}, \bar{p}, d_0, \bar{r}, \gamma \rangle$. We can define the corresponding value functions, and show they are connected to their classical counterparts.

$$\bar{v}_{\bar{\pi}}(s) \doteq \sum_{t=0}^{\infty} \mathbb{E}_{\bar{\pi}}\big[\gamma^t R_{t+1}|S_0 = s\big], \quad \bar{q}_{\bar{\pi}}(s, u) \doteq \mathbb{E}_{\bar{\pi}}\big[R_1 + \gamma \bar{v}_{\bar{\pi}}(S_1)|S_0 = s, U_0 = u\big]. \tag{9}$$

**Assumption 3.1.** The set $\mathcal{U}$ is continuous and compact, and the function $f(a|u)$ and its derivative are continuous with respect to $u$. When $\mathcal{S}$ or $\mathcal{A}$ is continuous, the corresponding set is assumed to be compact; moreover, the functions $p(s'|s, a)$, $d_0(s)$, $r(s, a)$, $\pi(s)$, $f(a|u)$, and their derivatives are also continuous with respect to $s$, $s'$, or $a$, respectively.

**Proposition 3.2.** *Under Assumption 3.1, $\bar{v}_{\bar{\pi}}(s) = v_{\pi}(s)$ and $\bar{q}_{\bar{\pi}}(s, u) = \mathbb{E}_{A \sim f(\cdot|u)}\big[q_{\pi}(s, A)\big]$.*

The proofs of Proposition 3.2 and all other theoretical results are presented in Appendix C.

The main advantage of this framework is that it transforms the original action space into a continuous parameter space $\mathcal{U}$, regardless of whether the underlying action space $\mathcal{A}$ is discrete, continuous,

or structured. This unification allows us to develop generic RL algorithms that operate over a continuous transformed action space, enabling a single framework to accommodate a wide variety of settings, including discrete-continuous hybrid action spaces (Masson et al., 2016). For example, we can apply DPG methods even in discrete action domains, where they were not previously applicable. We explore this direction in detail in Sections 4 and 5.

## 4 DISTRIBUTIONS-AS-ACTIONS POLICY GRADIENT ALGORITHMS

In this section, we introduce the *Distributions-as-Actions Policy Gradient* (DA-PG), a generalization of DPG for the distributions-as-actions framework. We show this estimator has lower variance, and then present a practical DA-PG algorithm for deep RL.

### 4.1 DISTRIBUTIONS-AS-ACTIONS POLICY GRADIENT ESTIMATOR

DA-PG is obtained by applying DPG to the distributions-as-actions MDP, as stated formally below.

**Assumption 4.1.** The function $\bar{\pi}_{\boldsymbol{\theta}}(s)$ and its derivative are continuous with respect to $\boldsymbol{\theta}$.

**Theorem 4.2** (Distributions-as-actions policy gradient theorem). *Under Assumptions 3.1 and 4.1, the gradient of the objective $J(\bar{\pi}_{\boldsymbol{\theta}}) = \sum_{t=0}^{\infty} \mathbb{E}_{\bar{\pi}}\left[\gamma^t R_{t+1}\right]$ with respect to $\boldsymbol{\theta}$ can be expressed as*

$$\nabla_{\boldsymbol{\theta}} J(\bar{\pi}_{\boldsymbol{\theta}}) = \mathbb{E}_{S \sim d_{\bar{\pi}_{\boldsymbol{\theta}}}}\left[\nabla_{\boldsymbol{\theta}} \bar{\pi}_{\boldsymbol{\theta}}(S)^{\top} \nabla_U \bar{q}_{\bar{\pi}_{\boldsymbol{\theta}}}(S, U)|_{U=\bar{\pi}_{\boldsymbol{\theta}}(S)}\right],$$

*where $d_{\bar{\pi}_{\boldsymbol{\theta}}}(s) \doteq \sum_{t=0}^{\infty} \mathbb{E}_{\bar{\pi}_{\boldsymbol{\theta}}}\left[\gamma^t \mathbb{I}(S_t = s)\right]$ is the (discounted) occupancy measure under $\bar{\pi}_{\boldsymbol{\theta}}$.*

The resulting gradient estimator of the surrogate objective $\hat{J}(\bar{\pi}_{\boldsymbol{\theta}}) = \mathbb{E}_{S_t \sim d}\left[\bar{Q}_{\mathbf{w}}(S_t, \bar{\pi}_{\boldsymbol{\theta}}(S_t))\right]$ is

$$\hat{\nabla}_{\boldsymbol{\theta}}^{\text{DA-PG}} \hat{J}(\bar{\pi}_{\boldsymbol{\theta}}; S_t) = \nabla_{\boldsymbol{\theta}} \bar{\pi}_{\boldsymbol{\theta}}(S_t)^{\top} \nabla_U \bar{Q}_{\mathbf{w}}(S_t, U)|_{U=\bar{\pi}_{\boldsymbol{\theta}}(S_t)}, \tag{10}$$

where $\bar{Q}_{\mathbf{w}}$ is a learned parameterized critic. Note that the DA-PG estimator shares the same mathematical form as the DPG estimator (Equation (6)). However, the roles of the components differ: in DA-PG, the policy $\bar{\pi}_{\boldsymbol{\theta}}$ outputs distribution parameters rather than a single action, and the critic estimates the expected return over the entire action distribution, rather than for a specific action.

In fact, DA-PG is a strict generalization of DPG. When the policy is restricted to be deterministic, the distribution parameters effectively become the action, and the distributions-as-actions critic reduces to the classical action-value critic.

**Proposition 4.3.** *If $\mathcal{U} = \mathcal{A}$ and $f(\cdot|u)$ is the Dirac delta distribution centered at $u$, then $\bar{\pi}_{\boldsymbol{\theta}}$ and $\bar{Q}_{\mathbf{w}}$ are equivalent to $\pi_{\boldsymbol{\theta}}$ and $Q_{\mathbf{w}}$, respectively. Consequently, the DA-PG gradient estimator becomes equivalent to DPG:*

$$\hat{\nabla}_{\boldsymbol{\theta}}^{\text{DA-PG}} \hat{J}(\bar{\pi}_{\boldsymbol{\theta}}; S_t) = \hat{\nabla}_{\boldsymbol{\theta}}^{\text{DPG}} \hat{J}(\pi_{\boldsymbol{\theta}}; S_t).$$

Moreover, DPG's theoretical analysis can also be extended to the distributions-as-actions framework. In Appendix C.3, we generalize the convergence analysis of DPG to DA-PG, establishing a theoretical guarantee that holds for MDPs with arbitrary action space types.

### 4.2 COMPARISON TO OTHER ESTIMATORS FOR STOCHASTIC POLICIES

We now compare the proposed DA-PG estimator with classical stochastic policy gradient methods, highlighting its variance and bias characteristics across action spaces.

DA-PG can be seen as the conditional expectation of both the LR (Equation (5)) and RP (Equation (7)) estimators. This leads to strictly lower variance for stochastic policies.

**Proposition 4.4.** *Assume $Q_{\mathbf{w}} = q_{\pi_{\boldsymbol{\theta}}}$ in $\hat{\nabla}_{\boldsymbol{\theta}}^{LR} \hat{J}(\pi_{\boldsymbol{\theta}}; S_t, A)$ and $\bar{Q}_{\mathbf{w}} = \bar{q}_{\bar{\pi}_{\boldsymbol{\theta}}}$ in $\hat{\nabla}_{\boldsymbol{\theta}}^{DA-PG} \hat{J}(\bar{\pi}_{\boldsymbol{\theta}}; S_t)$. Then, $\hat{\nabla}_{\boldsymbol{\theta}}^{DA-PG} \hat{J}(\bar{\pi}_{\boldsymbol{\theta}}; S_t) = \mathbb{E}_{A \sim \pi_{\boldsymbol{\theta}}(\cdot|S_t)}\left[\hat{\nabla}_{\boldsymbol{\theta}}^{LR} \hat{J}(\pi_{\boldsymbol{\theta}}; S_t, A)\right]$. Further, if the expectation of the action-conditioned variance is greater than zero, then $\mathbb{V}\left(\hat{\nabla}_{\boldsymbol{\theta}}^{DA-PG} \hat{J}(\bar{\pi}_{\boldsymbol{\theta}}; S_t)\right) < \mathbb{V}\left(\hat{\nabla}_{\boldsymbol{\theta}}^{LR} \hat{J}(\pi_{\boldsymbol{\theta}}; S_t, A)\right)$.*

**Proposition 4.5.** *Assume $\mathcal{A}$ is continuous, $Q_{\mathbf{w}} = q_{\pi_{\boldsymbol{\theta}}}$ in $\hat{\nabla}_{\boldsymbol{\theta}}^{RP} \hat{J}(\pi_{\boldsymbol{\theta}}; S_t, \epsilon)$, and $\bar{Q}_{\mathbf{w}} = \bar{q}_{\bar{\pi}_{\boldsymbol{\theta}}}$ in $\hat{\nabla}_{\boldsymbol{\theta}}^{DA-PG} \hat{J}(\bar{\pi}_{\boldsymbol{\theta}}; S_t)$. Then, $\hat{\nabla}_{\boldsymbol{\theta}}^{DA-PG} \hat{J}(\bar{\pi}_{\boldsymbol{\theta}}; S_t) = \mathbb{E}_{\epsilon \sim p}\left[\hat{\nabla}_{\boldsymbol{\theta}}^{RP} \hat{J}(\pi_{\boldsymbol{\theta}}; S_t, \epsilon)\right]$. Further, if the expectation of the noise-induced variance is greater than zero, then $\mathbb{V}\left(\hat{\nabla}_{\boldsymbol{\theta}}^{DA-PG} \hat{J}(\bar{\pi}_{\boldsymbol{\theta}}; S_t)\right) < \mathbb{V}\left(\hat{\nabla}_{\boldsymbol{\theta}}^{RP} \hat{J}(\pi_{\boldsymbol{\theta}}; S_t, \epsilon)\right)$.*

In discrete action spaces, the LR estimator typically requires carefully designed baselines to manage high variance, especially as dimensionality increases. While biased alternatives like the straight-through (ST) estimator (Bengison et al., 2013) or continuous relaxations (Jang et al., 2017; Maddison et al., 2017) exist, they sacrifice unbiasedness even when using a perfect critic. DA-PG avoids this trade-off, providing the first unbiased RP-style estimator with low variance in the discrete setting.

In continuous action spaces, DPG offers zero variance but assumes fixed stochasticity (i.e., no learnable exploration). RP estimators allow for learning the stochastic parameters but exhibit higher variance. DA-PG offers the best of both worlds: it permits learning all policy parameters including those for stochasticity while retaining the zero-variance property per state.

Another direction to reduce variance is *expected policy gradient* (EPG; Ciosek & Whiteson, 2018; Allen et al., 2017). The idea is to integrate (or sum) over actions, yielding zero-variance gradients conditioned on a state: $\hat{\nabla}_{\boldsymbol{\theta}}^{\text{EPG}} \hat{J}(\pi_{\boldsymbol{\theta}}; S_t) = \nabla_{\boldsymbol{\theta}} \mathbb{E}_{A_t \sim \pi_{\boldsymbol{\theta}}(\cdot|S_t)} [Q_{\mathbf{w}}(S_t, A_t)]$. However, this estimator is only practical in low-dimensional discrete action spaces (Allen et al., 2017) or in special cases within continuous settings—such as Gaussian policies with quadratic critics (Ciosek & Whiteson, 2020). In contrast, our estimator $\hat{\nabla}_{\boldsymbol{\theta}}^{\text{DA-PG}} \hat{J}(\bar{\pi}_{\boldsymbol{\theta}}; S_t)$ generalizes to a wider range of settings, including high-dimensional discrete, general continuous, and even hybrid action spaces.

Despite its lower variance, DA-PG may suffer from increased bias due to the increased complexity of the critic's input space. For discrete actions, the critic $\bar{Q}_{\mathbf{w}}$ inputs a vector of probabilities corresponding to discrete outcomes. For continuous actions, with Gaussian policies, the critic $\bar{Q}_{\mathbf{w}}$ inputs both the mean and standard deviation. This increased input dimensionality may make it harder to approximate the true action-value function, and if the critic is inaccurate, the overall benefit of lower gradient variance may be diminished—an effect we examine empirically in Section 5.5.

## 4.3 INTERPOLATED CRITIC LEARNING

In this section, we propose a method to improve learning the distributions-as-actions critic $\bar{Q}_{\mathbf{w}}$. Similar to Equation (3), the standard TD update for $\bar{Q}_{\mathbf{w}}$ is

$$\mathbf{w} \leftarrow \mathbf{w} + \alpha \big( R_{t+1} + \gamma \bar{Q}_{\mathbf{w}}(S_{t+1}, \bar{\pi}_{\boldsymbol{\theta}}(S_{t+1})) - \bar{Q}_{\mathbf{w}}(S_t, U_t) \big) \nabla_{\mathbf{w}} \bar{Q}_{\mathbf{w}}(S_t, U_t). \qquad (11)$$

This update, however, does not make use of the sampled action $A_t$ or its relationship to the resulting state and reward. If $A_t$ is made observable to the agent, the transition itself can be leveraged to update the value at alternative distribution parameters $\hat{U}_t$. This is possible because the observed action $A_t$ could have been sampled from distributions parameterized by many such $\hat{U}_t$. As a result, we can update the critic *off-distribution*, i.e., at distribution parameters different from those used to generate the transition. This parallels off-policy learning, but operates over action distributions rather than distinct policies.

What, then, should we choose for $\hat{U}_t$? To answer this, we ask: *what properties should the critic have to support effective policy optimization in parameter space?* Our answer is that the critic should provide informative gradient directions that guide the policy toward optimality. For MDPs, there always exists a deterministic optimal policy (Puterman, 2014). Therefore, we assume the existence of some $U_{A_t^*} \in \mathcal{U}$, a deterministic distribution corresponding to the optimal action $A_t^*$ for state $S_t$. Ideally, the critic should exhibit curvature that points toward such optimal parameters $U_t^*$.

One candidate for $\hat{U}_t$ is $U_{A_t}$, the deterministic distribution parameters associated with the sampled action $A_t$. However, merely learning accurate values at $U_{A_t}$ does not ensure that the critic has smooth curvature from $U_t$ toward high-value points. To encourage the critic to generalize better and provide smoother gradients, we propose using a linearly interpolated point between $U_t$ and $U_{A_t}$:

$$\hat{U}_t = \omega_t U_t + (1 - \omega_t) U_{A_t}, \quad \omega_t \sim \text{Uniform}[0, 1]. \qquad (12)$$

The critic is trained to predict the value at $\hat{U}_t$ using the following update:

$$\mathbf{w} \leftarrow \mathbf{w} + \alpha \big( R_{t+1} + \gamma \bar{Q}_{\mathbf{w}}(S_{t+1}, \bar{\pi}_{\boldsymbol{\theta}}(S_{t+1})) - \bar{Q}_{\mathbf{w}}(S_t, \hat{U}_t) \big) \nabla_{\mathbf{w}} \bar{Q}_{\mathbf{w}}(S_t, \hat{U}_t). \qquad (13)$$

We refer to this approach as *Interpolated Critic Learning* (ICL). Note that the ICL update omits *importance sampling* weights that correct for the mismatch between $f(\cdot|U_t)$ and $f(\cdot|\hat{U}_t)$, and is therefore generally biased. For simplicity, we do not include such corrections and leave a principled treatment of importance weighting and more advanced off-distribution updates to future work.

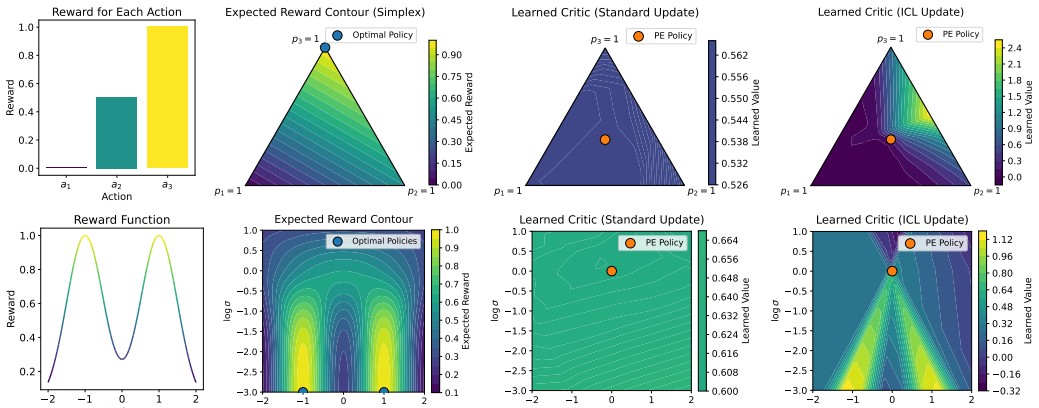

Figure 2: Visualization of the **reward function** (col 1), **expected rewards of distribution parameters** (col 2), and **learned critics** using the *standard* update in Equation (11) (col 3) and the *Interpolated Critic Learning* (ICL) update in Equation (13) (col 4) in policy evaluation (PE). **Top:** K-Armed Bandit. **Bottom:** Bimodal Continuous Bandit. With access only to samples from the PE policy (the fixed policy being evaluated), the standard update estimates values accurately at that policy but fails to generalize beyond it. In contrast, the ICL update learns a critic that captures curvature information useful for policy optimization.

To further provide intuition on ICL, we conduct a policy evaluation experiment in bandit problems, shown in Figure 2 (column 1). Figure 2 (column 3) and (column 4) show the learned critics using the standard update in Equation (11) and the ICL update in Equation (13), respectively. The critic learned by ICL has more informative curvature. As a result, the policy can be updated toward high-value regions more easily. In the continuous action case, the learned critic is sufficient to update the policy toward near-optimal distribution parameters. More details can be found in Appendix D.2.

### 4.4 Distributions-as-actions actor-critic

Since DA-PG is derived from DPG, we base our practical algorithm on TD3 (Fujimoto et al., 2018), a strong DPG-based off-policy actor-critic algorithm for continuous control. We replace the classical actor and critic with their distributions-as-actions counterparts and use the DA-PG estimator (Equation (10)) and the ICL critic loss (Equation (13)) to update them, respectively. We omit the actor target network, as it does not improve performance (see Appendix D.4). The pseudocode for the resulting algorithm, *Distributions-as-Actions Actor-Critic* (DA-AC), is provided in Appendix G.

## 5 Experiments

In this section, we conduct experiments to investigate DA-AC's empirical performance in continuous (Section 5.1), discrete (Sections 5.2 and 5.3), and hybrid (Section 5.4) control settings. In addition, we examine the effectiveness of the proposed interpolated critic learning in Section 5.5. Unless otherwise noted, each environment is run with 10 seeds, and error bars or shaded regions indicate 95% bootstrap confidence intervals.

### 5.1 Continuous control

We use OpenAI Gym MuJoCo (Brockman et al., 2016) and the DeepMind Control (DMC) Suite (Tunyasuvunakool et al., 2020) for continuous control. From MuJoCo, we use the commonly used 5 environments: Hopper-v3, Walker2d-v3, HalfCheetah-v3, Ant-v3, and Humanoid-v3; from DMC, we use the same 15 environments as D'Oro et al. (2023). Details about these environments are in Appendix D.4. We run for 1 million ($1M$) steps in each environment.

**Algorithms** We use TD3 (Fujimoto et al., 2018) as our primary baseline, as DA-AC is based on it. We also include an off-policy actor-critic baseline that uses the reparameterization (RP) estimator. This RP-AC algorithm closely resembles DA-AC but learns in the original action space and updates

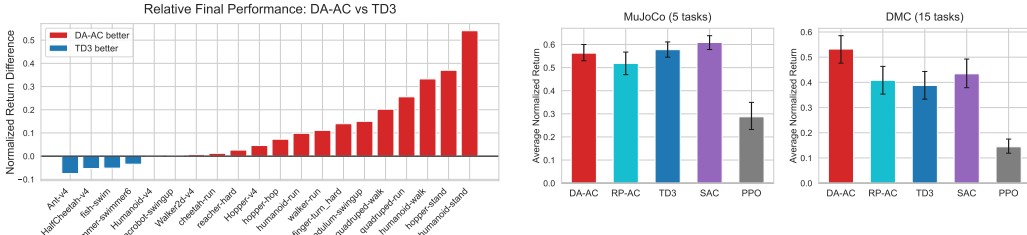

Figure 3: **Relative final performance of DA-AC versus TD3** across 20 individual *continuous* control tasks (col 1), and **average normalized returns of DA-AC and baselines** on MuJoCo (col 2) and DeepMind Control (col 3) tasks. In individual task comparisons (col 1), results are averaged over 10 seeds per task. For average performance plots (cols 2-3), values are averaged over 10 seeds and tasks. Error bars show 95% bootstrap confidence intervals (CIs).

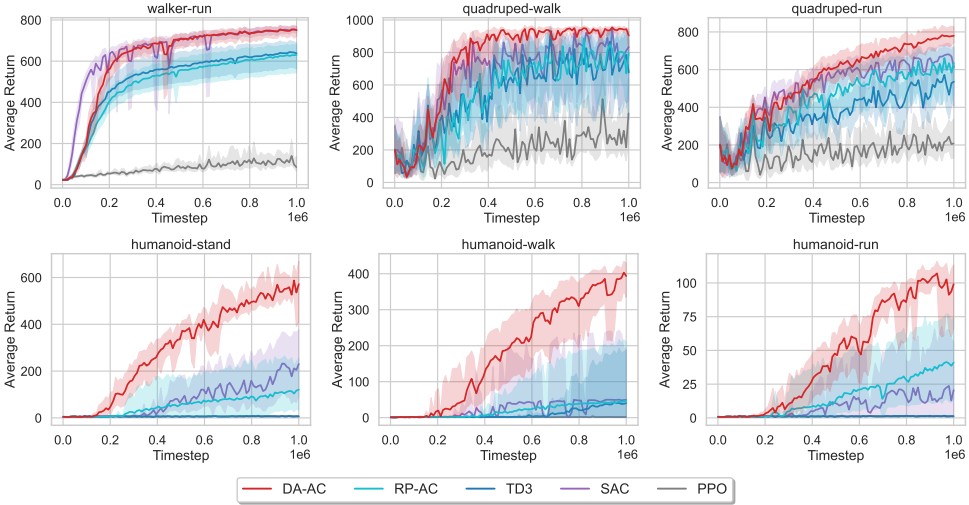

Figure 4: **Learning curves in six DeepMind Control tasks with high-dimensional action spaces.** Results are averaged over 10 seeds. Shaded regions show 95% bootstrap CIs.

the policy using the RP estimator. For consistency, DA-AC and RP-AC use the default hyperparameters of TD3 and a Gaussian policy parameterization. Implementations details and pseudocode can be found in Appendices D.4 and G, respectively. For reference, we also evaluate the performance of SAC (Haarnoja et al., 2018) and PPO (Schulman et al., 2017). We focus on the single-stream learning setting, so we use the original single-stream version of PPO as in Schulman et al. (2017).

**Results**  Figure 3 shows per-environment performance for DA-AC and TD3 and the aggregated results across environments for all algorithms. From Figure 3 (column 1), we can see that DA-AC achieves better performance in more environments compared to TD3. From Figure 3 (columns 2–3), we can see that DA-AC achieves better overall performance, outperforming most baselines significantly in the DMC Suite, particularly in high-dimensional environments (see Figure 4).

## 5.2 DISCRETE CONTROL

Following Ceron & Castro (2021), we use 4 Gym classic control (Brockman et al., 2016) and 5 MinAtar environments (Young & Tian, 2019) for discrete control. We run each environment for $500k$ (classic control) or $5M$ (MinAtar) steps.

**Algorithms**  We include off-policy actor-critic baselines that resemble DA-AC. These baselines learn in the original action space and update the policy with different gradient estimators, including the likelihood-ratio (LR-AC) and expected (EAC) policy gradient estimators. Here, LR-AC uses a state-value baseline analytically computed from action values. Although not common in prior work, we also include a variant that uses the straight-through (ST) estimator (Bengio et al., 2013), denoted

Figure 5: **Average normalized returns of DA-AC and baselines** on *discrete* control benchmarks, including classic control (col 1), MinAtar (col 2), discretized MuJoCo (col 3), and discretized Deep-Mind Control (col 4) tasks.

as ST-AC. This baseline is the discrete counterpart of RP-AC, serving as a performance reference for alternative RP-based methods. For comparison, we also evaluate the performance of Discrete SAC (DSAC; Christodoulou, 2019), DQN (Mnih et al., 2015), and PPO (Schulman et al., 2017). The hyperparameters for DA-AC and X-AC baselines (LR-AC, ST-AC, and EAC) are adopted from the TD3 defaults and adjusted to the corresponding benchmark based on those of DQN. More details and pseudocode can be found in Appendices D.5 and G, respectively.

**Results**    From Figure 5 (columns 1–2), we can see that DA-AC is among the best-performing algorithms in both classic control and MinAtar, achieving comparable performance to DQN.

### 5.3 High-dimensional discrete control

We also conduct experiments on discrete control with high-dimensional action spaces. For this setting, we use the same 20 environment from Section 5.1 but with a discretized action space. Specifically, we discretize each action dimension into 7 bins with uniform spacing. For example, the original action space in Humanoid-v4 is $[-0.4, 0.4]^{17}$, which is discretized to $0.4 \times \{-1, -\frac{2}{3}, -\frac{1}{3}, 0, \frac{1}{3}, \frac{2}{3}, 1\}^{17}$. We run each environment for $1M$ steps.

**Algorithms**    We use ST-AC, LR-AC, and PPO from the previous section as baselines. EAC, DSAC, and DQN are excluded, as they are computationally infeasible in environments with high-dimensional discrete actions. In particular, DSAC—like EAC—relies on computing expected updates over the full action set; without the expected updates, it fails to learn. DQN would likewise require representing and maximizing over this action space, which is intractable. LR-AC learns an additional state-value function as a baseline, since analytically deriving it from the action-value function is prohibitive in this high-dimensional setting. We use the same hyperparameters as those in Section 5.1. More details can be found in Appendix D.6.

**Results**    As shown in Figure 5 (columns 3–4), DA-AC's average performance is higher than all baselines in both benchmarks. Note that the performance of DA-AC and PPO is comparable to the original continuous action setting (see columns 2–3 in Figure 3). Figure 12 in Appendix D.6 also contrasts the performance of DA-AC with discrete versus continuous actions.

### 5.4 Hybrid control

In addition to continuous and discrete control settings, we also evaluate DA-AC's performance in parameterized-action MDPs (PAMDPs), a hybrid control setting with parameterized actions (see Masson et al. (2016) for detailed discussion). We use 7 PAMDP environments from Li et al. (2022) and follow their experiment protocol. See Appendix D.7 for more details.

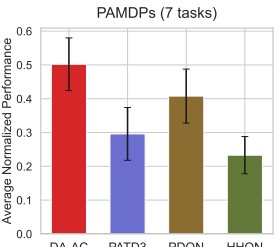

**Algorithms**    We use PATD3 as our primary baseline, a DPG-based baseline specifically designed for parameterized action (PA) spaces. PATD3 builds on PADDPG (Hausknecht & Stone, 2016) and incorporates clipped double Q-learning from TD3, making it a suitable and directly comparable baseline for DA-AC, as both methods build on TD3. In DA-AC, the distribution

Figure 6: **Average normalized performance of DA-AC and baselines** on *hybrid* control tasks.

parameters include both the probability vector for the discrete actions and mean/log-std vectors for

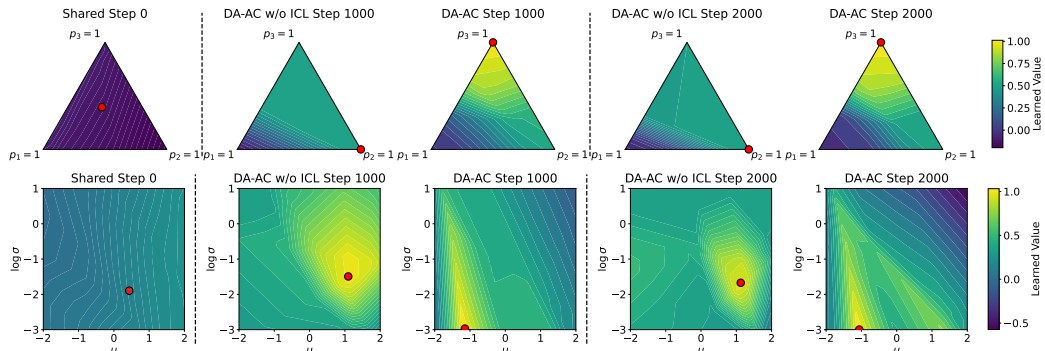

Figure 8: **Initial critic** (col 1) and **learned critics** at different training stages for DA-AC w/o ICL (cols 2 and 4) and DA-AC (cols 3 and 5). **Top:** K-Armed Bandit. **Bottom:** Bimodal Continuous Bandit. Red dots indicate the current policy parameters. DA-AC produces more accurate value estimates at deterministic distribution parameters—corresponding to the vertices in the discrete case and the x-axis in the continuous case—and offers stronger gradient signals for policy optimization.

the continuous actions. We keep most hyperparameters the same as TD3's default unless otherwise adjusted to align with PATD3. In addition, we also include PDQN (Xiong et al., 2018) and HHQN (Fu et al., 2019) as additional baselines for reference. See Appendix D.7 for more details.

**Results** Figure 6 shows the average normalized performance of DA-AC and baselines. The learning curves in each individual environment can be found in Figure 16. We can see that DA-AC also achieves competitive or better performance than the baselines.

## 5.5 EFFECTIVENESS OF INTERPOLATED CRITIC LEARNING

We compare DA-AC and DA-AC w/o ICL, an ablated version that uses the standard critic update (Equation (11)). As shown in Figure 7, DA-AC w/o ICL generally underperforms DA-AC across settings; the paired $t$-test in Appendix D.8 similarly indicates a statistically significant positive mean improvement.

To provide further insights into the differences, we move to a bandit setting where visualization and analysis are intuitive. We use the same bandit environments from Figure 2, and run each algorithm for 2000 steps and 50 seeds. See Appendix D.3 for hyperparameters and details.

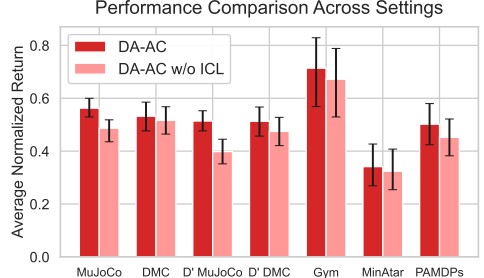

Figure 7: **Comparison between DA-AC and DA-AC w/o ICL** in all settings.

Figure 9 in the appendix shows the superiority of DA-AC over DA-AC w/o ICL, as well as the bias-variance trade-off incurred by different gradient estimators. To assess the impact of ICL on critic quality, we visualize the learned critics from a representative training run of DA-AC and DA-AC w/o ICL in Figure 8. In both discrete and continuous action settings, DA-AC yields a significantly improved critic landscape early in training.

## 6 CONCLUSIONS

We introduced the *distributions-as-actions framework*, redefining the agent-environment boundary to treat distribution parameters as actions. We showed that the policy gradient update has theoretically lower variance, and developed a practical deep RL algorithm called *Distributions-as-Actions Actor-Critic* (DA-AC) based on this estimator. We also introduced an improved critic learning update, ICL, tailored to this new setting. We demonstrated that DA-AC achieves competitive performance in diverse settings across continuous, discrete, and hybrid control.

This reframing allowed us to develop a continuous action algorithm that applies to diverse underlying action types. A key next step is to further exploit this reframing for new algorithmic avenues, including model-based methods, hierarchical control, or novel hybrid approaches. See Appendix B for a discussion of how existing RL algorithms can be extended within this framework. There are also key open questions around critic learning. More advanced strategies for training the distributions-as-actions critic could be investigated, including off-distribution updates at diverse regions of the parameter space or using a learned action-value function $Q_{\mathbf{w}}(s, a)$ to guide updates of $\bar{Q}_{\mathbf{w}'}(s, u)$. This will also open up new questions about convergence properties for these new variants.

### ACKNOWLEDGMENTS

This research is supported by the Canada CIFAR AI Chairs program, the Reinforcement Learning and Artificial Intelligence (RLAI) laboratory, the Alberta Machine Intelligence Institute (Amii), and the Natural Sciences and Engineering Research Council (NSERC) of Canada. Jiamin He also gratefully acknowledges Richard S. Sutton, Csaba Szepesvári, Bryan Chan, and Shivam Garg for valuable discussions and thanks the Digital Research Alliance of Canada for providing computational resources.

### REPRODUCIBILITY STATEMENT

To facilitate reproducibility, we provide a code release covering DA-AC implementations across all control settings at `https://github.com/hejm37/da-ac`. Comprehensive hyperparameter choices and environment configurations are documented in Appendix D. All reported metrics are based on multiple seeds, with uncertainty quantified using $95\%$ bootstrap confidence intervals.

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

## A  RELATED WORKS

In this section, we provide an extended discussion of related work.

**Value-based control**     When the action space is discrete, value-based algorithms are one of the most commonly used approaches (Watkins & Dayan, 1992; Mnih et al., 2015; van Hasselt et al., 2016; Hessel et al., 2018). By learning an action-value function, these algorithms extract policies using various greedy operators. While these methods have been effective in a wide range of discrete control domains, their applications to continuous-action problems are limited, with only a few exceptions (Seyde et al., 2023).

**Policy-based discrete control**     Policy-based methods, including actor-critic algorithms, form another important class of approaches for discrete control (Williams, 1992; Mnih et al., 2016). These methods explicitly maintain a policy that outputs distribution parameters used to construct a policy distribution from which actions are sampled. In the discrete case, these parameters correspond to the logits of a categorical distribution. Beyond the likelihood-ratio (LR) policy gradient estimator (Williams, 1992), straight-through (ST; Bengio et al., 2013), Gumbel-Softmax (Jang et al., 2017), and Concrete (Maddison et al., 2017) estimators provide reparameterization-based but biased gradient estimation. In contrast to these biased estimators, our distributions-as-actions (DA) gradient estimator in Equation (10) can be viewed as the first unbiased reparameterization (RP) estimator for discrete distributions.

**Policy-based continuous control**     For continuous control, policy-based methods dominate the literature (van Hasselt & Wiering, 2007; Silver et al., 2014; Lillicrap et al., 2016; Schulman et al., 2017; Haarnoja et al., 2018). The policy typically outputs the parameters of a parametric distribution. Gaussian policies are the most common choice (Lillicrap et al., 2016; Schulman et al., 2017; Haarnoja et al., 2018), although many alternatives have been explored in different contexts (Chou et al., 2017; Bedi et al., 2024; Zhu et al., 2025; He et al., 2025). Optimizing these policies using classical policy gradient estimators (LR or RP) requires access either to an analytical log-density function or a reparameterization function. In contrast, the DA gradient estimator in Equation (10) requires neither, enabling application to a broader class of policies. Beyond parametric distributions, implicit policies built using more expressive generative models have also been studied (Haarnoja et al., 2017; Messaoud et al., 2024). Our DA framework and estimator can be applied to these advanced policy classes as well, which suggests an interesting direction for future work.

**Policy-based continuous control with discretization**     Another line of work discretizes the continuous action space and then applies discrete-action control algorithms—often policy-based methods due to the high dimensionality of action spaces (Tang & Agrawal, 2020; Seyde et al., 2021; Zhu et al., 2024). While such approaches have shown strong benchmark performance, they may be undesirable in practice because the resulting control can be less smooth and more unstable (Seyde et al., 2021), and the method often requires additional tuning of the discretization granularity (Tang & Agrawal, 2020). In this work, we treat discretized continuous control problems primarily as a testbed for high-dimensional discrete control. Thus, we do not extensively compare continuous-vs. discrete-based methods for continuous control, as this is not our main focus. Nevertheless, we include such a comparison in Figure 12 for reference.

**Policy-based hybrid control**     Beyond purely discrete or continuous settings, many real-world applications involve hybrid action spaces requiring the agent to control discrete and continuous variables simultaneously (Masson et al., 2016; Xiong et al., 2018). These problems can often be modeled as parameterized action MDPs (PAMDPs; Masson et al., 2016), in which the agent selects a discrete action and its associated continuous parameters. Most standard discrete and continuous control algorithms are not directly applicable to PAMDPs and require additional adaptation or hybridization to handle such action structures. For example, DDPG (Lillicrap et al., 2016) and PPO (Schulman et al., 2017) have been modified to support hybrid actions (Hausknecht & Stone, 2016; Fan et al., 2019), and combinations of DDPG and DQN (Mnih et al., 2015) have also been explored (Xiong et al., 2018; Fu et al., 2019). Unlike these methods, which patch together or retrofit existing algorithms, our DA reframing directly converts hybrid control into a continuous control problem, enabling a simple, unified algorithm applicable to PAMDPs and even more general hybrid settings.

**Representation-driven RL**     Different from the traditional policy optimization perspective used in the methods above, representation-driven RL (RepRL) offers an alternative viewpoint (Nabati et al., 2023). Instead of optimizing policy parameters $\boldsymbol{\theta}$ by estimating gradients based on the values of sampled state-action pairs, RepRL recasts the search for optimal $\boldsymbol{\theta}$ as a linear bandit problem by projecting $\boldsymbol{\theta}$ into a lower-dimensional representation $f(\boldsymbol{\theta})$ and optimizing $\boldsymbol{\theta}$ based on its expected value over states. While our proposed DA framework also redraws the decision boundary, it does so in a fundamentally different way. RepRL retreats the decision boundary all the way to a bandit problem and, in some sense, treats even the policy network $\pi$ as part of the environment. In contrast, the DA framework only reframes the distribution parameters themselves as the decision variables, viewing only the sampling function as part of the environment.

**D-MDP and dynamical system optimization**     Concurrently, Todorov (2025b) studied D-MDP, which is closely related to the distributions-as-actions (DA) MDP formulation considered in our work. Their primary focus is on using D-MDP to establish connections between stochastic and deterministic policy gradients in continuous action settings. In contrast, we emphasize the applicability of the DA-MDP formulation to diverse action spaces and develop practical algorithms under this perspective. While their work concentrates on theoretical characterization, we complement this viewpoint with empirical investigation, demonstrating the viability of DA-MDP-based algorithms across a variety of action space types. In subsequent work, Todorov (2025a) further proposed a dynamical system optimization perspective that connects a broader class of problem settings. Exploring the relationship between our proposed algorithm and this dynamical systems view could be an interesting direction for future work.

**On the agent-environment boundary**     More broadly, the agent-environment boundary has also been explored from several other angles (Jiang, 2019; Harutyunyan, 2020; Abel et al., 2025). While Harutyunyan (2020) and Abel et al. (2025) focus on the implications of the boundary from a philosophical perspective, Jiang (2019) discusses how theoretical results might be impacted when the agent's state space is defined differently. In contrast to these works, our work focuses specifically on a shift in the agent's action space and the algorithmic consequences.

# B  EXTENDING EXISTING RL ALGORITHMS TO THE DA FRAMEWORK

While we have explored only one model-free actor-critic algorithm under the proposed distributions-as-actions (DA) framework, many other algorithms in the classical RL literature can also be extended to this setting. To facilitate future research, we outline several such directions below.

**Entropy regularization**     Entropy regularization is a widely used mechanism for encouraging exploration in RL, and incorporating it into the DA framework represents a promising avenue. We discuss two potential approaches for adding entropy regularization to DA-AC. The first approach is to augment the policy optimization objective (Equation (4)) with an entropy term. This requires adding an entropy component to the policy gradient estimator (Equation (10)):

$$\hat{\nabla}_{\boldsymbol{\theta}}^{\text{DA-PG}} \hat{J}(\bar{\pi}_{\boldsymbol{\theta}}; S_t) = \nabla_{\boldsymbol{\theta}} \bar{\pi}_{\boldsymbol{\theta}}(S_t)^{\top} \nabla_U \bar{Q}_{\mathbf{w}}(S_t, U)\big|_{U=\bar{\pi}_{\boldsymbol{\theta}}(S_t)} + \alpha \nabla_{\boldsymbol{\theta}} \mathcal{H}(f(\cdot|\bar{\pi}_{\boldsymbol{\theta}}(S_t))), \quad (14)$$

where $\alpha$ is the entropy coefficient and $\mathcal{H}(f)$ denotes the entropy of distribution $f$. Optionally, the critic may also incorporate entropy, yielding the Maximum Entropy RL formulation (MaxEnt; Haarnoja et al., 2018). The standard MaxEnt critic update within the DA framework becomes

$$\mathbf{w} \leftarrow \mathbf{w} + \alpha\Big(R_{t+1} + \gamma\big(\bar{Q}_{\mathbf{w}}(S_{t+1}, \bar{\pi}_{\boldsymbol{\theta}}(S_{t+1})) + \alpha \mathcal{H}(f(\cdot|\bar{\pi}_{\boldsymbol{\theta}}(S_{t+1})))\big) - \bar{Q}_{\mathbf{w}}(S_t, U_t)\Big)\nabla_{\mathbf{w}}\bar{Q}_{\mathbf{w}}(S_t, U_t).$$
$$(15)$$

The second approach is specific to the MaxEnt setting and incorporates entropy directly into the reward:

$$R'_{t+1} = R_{t+1} + \alpha \mathcal{H}(f(\cdot|U_t)). \quad (16)$$

Under this reward shaping, the optimization problem coincides with the MaxEnt objective (Haarnoja et al., 2018). This method does not require modifying the actor or critic updates; only the reward is transformed. The entropy can be computed analytically when available or estimated via samples (e.g., using $-\log f(A_t|U_t)$). Understanding the trade-offs between these alternatives is itself an interesting open question.

**Model-based planning algorithms** Beyond model-free methods, model-based planning algorithms can also be incorporated into the DA framework. A straightforward approach is to combine traditional model-based algorithms operating on primitive actions with DA-based value estimation. For example, in discrete-action environments, one can apply Monte Carlo tree search (MCTS; Silver et al., 2016) over the primitive discrete actions while using DA for the critic.

A potentially more compelling direction is to learn a model over the distribution parameters themselves. This would make it possible to apply continuous-action model-based planners—such as continuous-action variants of MCTS (Yee et al., 2016), TD-MPC (Hansen et al., 2022), or model-based reparameterization gradient methods (Zhang et al., 2023)—directly within the DA framework.

**Incompatibility with discrete-structure-based algorithms** By treating distributions as actions, we might lose the ability to exploit certain convenient structures of the original action space—particularly in discrete settings. While this choice allows the DA framework to remain agnostic to the specifics of the primitive action space, it may still be desirable to leverage action structure when beneficial. Hybrid approaches that combine the DA framework with structure-aware algorithms, such as integrating MCTS with DA-based value estimation, provide one promising path forward.

## C   THEORETICAL ANALYSIS

We provide proofs of the theoretical results for the distributions-as-actions framework and Distributions-as-Actions Policy Gradient (DA-PG) in the main text in Appendices C.1 and C.2. In addition, we also extend a convergence proof of DPG from Xiong et al. (2022) to DA-PG in Appendix C.3.

### C.1   PROOFS OF THEORETICAL RESULTS IN SECTION 3

**Assumption 3.1.** The set $\mathcal{U}$ is continuous and compact, and the function $f(a|u)$ and its derivative are continuous with respect to $u$. When $\mathcal{S}$ or $\mathcal{A}$ is continuous, the corresponding set is assumed to be compact; moreover, the functions $p(s'|s, a)$, $d_0(s)$, $r(s, a)$, $\bar{\pi}(s)$, $f(a|u)$, and their derivatives are also continuous with respect to $s$, $s'$, or $a$, respectively.

**Proposition 3.2.** *Under Assumption 3.1, $\bar{v}_{\bar{\pi}}(s) = v_\pi(s)$ and $\bar{q}_{\bar{\pi}}(s, u) = \mathbb{E}_{A \sim f(\cdot|u)}\big[q_\pi(s, A)\big]$.*

*Proof.* Let $\pi$ be the policy in the original MDP that first maps $s$ to $u = \bar{\pi}(s)$ and then samples $A \sim f(\cdot|u)$. The state-value function $\bar{v}_{\bar{\pi}}(s)$ in the distributions-as-actions MDP is defined as

$$\bar{v}_{\bar{\pi}}(s) = \sum_{k=0}^{\infty} \mathbb{E}_{\bar{\pi}}\big[\gamma^k \bar{r}(S_k, U_k)|S_0 = s\big],$$

where $U_k = \bar{\pi}(S_k)$. From Equation (8), $\bar{r}(s, u) = \mathbb{E}_{A \sim f(\cdot|u)}\big[r(s, A)\big]$. Also, the transition $\bar{p}(s'|s, u) = \mathbb{E}_{A \sim f(\cdot|u)}\big[p(s'|s, A)\big]$. Consider a trajectory $S_0, U_0, S_1, U_1, \ldots$ in the distributions-as-actions MDP (DA-MDP). This corresponds to a trajectory $S_0, A_0, S_1, A_1, \ldots$ in the original MDP where $A_k \sim f(\cdot|U_k)$. The expected reward at time $k$ in the DA-MDP, given $S_k$ and $U_k = \bar{\pi}(S_k)$, is $\bar{r}(S_k, \bar{\pi}(S_k)) = \mathbb{E}_{A_k \sim f(\cdot|\bar{\pi}(S_k))}\big[r(S_k, A_k)\big]$. The dynamics are also equivalent in expectation: For example, when $\mathcal{S}$ is continuous, $\mathbb{E}\big[S_{k+1}|S_k, U_k\big] = \mathbb{E}_{S' \sim \bar{p}(\cdot|S_k, U_k)}\big[S'\big] = \mathbb{E}_{A_k \sim f(\cdot|U_k)}\big[\mathbb{E}_{S' \sim p(\cdot|S_k, A_k)}\big[S'\big]\big]$. Thus, the sequence of states and expected rewards generated under $\bar{\pi}$ in the DA-MDP is identical in distribution to the sequence of states and rewards under $\pi$ in the original MDP. Therefore, $\bar{v}_{\bar{\pi}}(s) = v_\pi(s)$.

For the action-value function $\bar{q}_{\bar{\pi}}(s, u)$,

$$\begin{aligned}
\bar{q}_{\bar{\pi}}(s, u) &= \mathbb{E}_{\bar{\pi}}\big[\bar{r}(S_0, U_0) + \gamma \bar{v}_{\bar{\pi}}(S_1)|S_0 = s, U_0 = u\big] \\
&= \bar{r}(s, u) + \gamma \mathbb{E}_{S_1 \sim \bar{p}(\cdot|s, u)}\big[\bar{v}_{\bar{\pi}}(S_1)\big] \\
&= \mathbb{E}_{A \sim f(\cdot|u)}\big[r(s, A)\big] + \gamma \mathbb{E}_{A \sim f(\cdot|u)}\big[\mathbb{E}_{S_1 \sim p(\cdot|s, A)}\big[v_\pi(S_1)\big]\big] \quad \text{(using } \bar{v}_{\bar{\pi}} = v_\pi) \\
&= \mathbb{E}_{A \sim f(\cdot|u)}\big[r(s, A) + \gamma \mathbb{E}_{S_1 \sim p(\cdot|s, A)}\big[v_\pi(S_1)\big]\big] \\
&= \mathbb{E}_{A \sim f(\cdot|u)}\big[\mathbb{E}_\pi\big[R_1 + \gamma v_\pi(S_1)|S_0 = s, A_0 = A\big]\big] \\
&= \mathbb{E}_{A \sim f(\cdot|u)}\big[q_\pi(s, A)\big].
\end{aligned}$$

The compactness and continuity assumptions in Assumption 3.1 ensure these expectations and value functions are well-defined. □

## C.2 Proofs of theoretical results in Section 4

**Assumption 4.1.** The function $\bar{\pi}_{\boldsymbol{\theta}}(s)$ and its derivative are continuous with respect to $\boldsymbol{\theta}$.

**Theorem 4.2** (Distributions-as-actions policy gradient theorem). *Under Assumptions 3.1 and 4.1, the gradient of the objective $J(\bar{\pi}_{\boldsymbol{\theta}}) = \sum_{t=0}^{\infty} \mathbb{E}_{\bar{\pi}}\left[\gamma^t R_{t+1}\right]$ with respect to $\boldsymbol{\theta}$ can be expressed as*

$$\nabla_{\boldsymbol{\theta}} J(\bar{\pi}_{\boldsymbol{\theta}}) = \mathbb{E}_{S \sim d_{\bar{\pi}_{\boldsymbol{\theta}}}}\left[\nabla_{\boldsymbol{\theta}} \bar{\pi}_{\boldsymbol{\theta}}(S)^{\top} \nabla_U \bar{q}_{\bar{\pi}_{\boldsymbol{\theta}}}(S, U)|_{U = \bar{\pi}_{\boldsymbol{\theta}}(S)}\right],$$

*where $d_{\bar{\pi}_{\boldsymbol{\theta}}}(s) \doteq \sum_{t=0}^{\infty} \mathbb{E}_{\bar{\pi}_{\boldsymbol{\theta}}}\left[\gamma^t \mathbb{I}(S_t = s)\right]$ is the (discounted) occupancy measure under $\bar{\pi}_{\boldsymbol{\theta}}$.*

*Proof.* This theorem results from applying the deterministic policy gradient (DPG) theorem to the DA-MDP $\langle \mathcal{S}, \mathcal{U}, \bar{p}, d_0, \bar{r}, \gamma \rangle$, where $\bar{\pi}_{\boldsymbol{\theta}} : \mathcal{S} \to \mathcal{U}$ acts as a deterministic policy. The objective function is $J(\bar{\pi}_{\boldsymbol{\theta}}) = \mathbb{E}_{S_0 \sim d_0}\left[\bar{v}_{\bar{\pi}_{\boldsymbol{\theta}}}(S_0)\right]$.

Following the DPG theorem derivation (Silver et al. (2014), Theorem 1), for a general deterministic policy $\mu_{\boldsymbol{\theta}} : \mathcal{S} \to \mathcal{A}$, the policy gradient is

$$\nabla_{\boldsymbol{\theta}} J(\mu_{\boldsymbol{\theta}}) = \mathbb{E}_{S \sim d_{\mu_{\boldsymbol{\theta}}}}\left[\nabla_{\boldsymbol{\theta}} \mu_{\boldsymbol{\theta}}(S)^{\top} \nabla_A q_{\mu_{\boldsymbol{\theta}}}(S, A)|_{A = \mu_{\boldsymbol{\theta}}(S)}\right].$$

In our context:

- The policy in the DA-MDP is $\bar{\pi}_{\boldsymbol{\theta}}(s)$.

- The action space is $\mathcal{U}$, and actions are denoted by $u$.

- The critic $\bar{q}_{\bar{\pi}_{\boldsymbol{\theta}}}(s, u)$ is the action-value function in this DA-MDP.

- The state distribution $d_{\bar{\pi}_{\boldsymbol{\theta}}}(s)$ is the discounted state occupancy measure under policy $\bar{\pi}_{\boldsymbol{\theta}}$.

Assumptions 3.1 and 4.1 ensure that $\bar{\pi}_{\boldsymbol{\theta}}(s)$ and $\bar{q}_{\bar{\pi}_{\boldsymbol{\theta}}}(s, u)$ are appropriately differentiable and that the interchange of expectation and differentiation is valid. Substituting $\bar{\pi}_{\boldsymbol{\theta}}$ for $\mu_{\boldsymbol{\theta}}$ and $\bar{q}_{\bar{\pi}_{\boldsymbol{\theta}}}$ for $q_{\mu_{\boldsymbol{\theta}}}$ yields the theorem's result:

$$\nabla_{\boldsymbol{\theta}} J(\bar{\pi}_{\boldsymbol{\theta}}) = \mathbb{E}_{S \sim d_{\bar{\pi}_{\boldsymbol{\theta}}}}\left[\nabla_{\boldsymbol{\theta}} \bar{\pi}_{\boldsymbol{\theta}}(S)^{\top} \nabla_U \bar{q}_{\bar{\pi}_{\boldsymbol{\theta}}}(S, U)|_{U = \bar{\pi}_{\boldsymbol{\theta}}(S)}\right].$$

The notation $\nabla_{\boldsymbol{\theta}} \bar{\pi}_{\boldsymbol{\theta}}(s)^{\top} \nabla_u \bar{q}_{\bar{\pi}_{\boldsymbol{\theta}}}$ in the theorem statement implies the appropriate vector or matrix product. If $\boldsymbol{\theta} \in \mathbb{R}^k$ and $u \in \mathbb{R}^m$, then $\nabla_{\boldsymbol{\theta}} \bar{\pi}_{\boldsymbol{\theta}}(s)$ is an $m \times k$ Jacobian, $\nabla_u \bar{q}_{\bar{\pi}_{\boldsymbol{\theta}}}$ is an $m \times 1$ vector, and the product $(\nabla_{\boldsymbol{\theta}} \bar{\pi}_{\boldsymbol{\theta}}(s))^{\top} \nabla_u \bar{q}_{\bar{\pi}_{\boldsymbol{\theta}}}$ results in the $k \times 1$ gradient vector for $J(\bar{\pi}_{\boldsymbol{\theta}})$. □

**Proposition 4.3.** *If $\mathcal{U} = \mathcal{A}$ and $f(\cdot|u)$ is the Dirac delta distribution centered at $u$, then $\bar{\pi}_{\boldsymbol{\theta}}$ and $\bar{Q}_{\mathbf{w}}$ are equivalent to $\pi_{\boldsymbol{\theta}}$ and $Q_{\mathbf{w}}$, respectively. Consequently, the DA-PG gradient estimator becomes equivalent to DPG:*

$$\hat{\nabla}_{\boldsymbol{\theta}}^{DA\text{-}PG} \hat{J}(\bar{\pi}_{\boldsymbol{\theta}}; S_t) = \hat{\nabla}_{\boldsymbol{\theta}}^{DPG} \hat{J}(\pi_{\boldsymbol{\theta}}; S_t).$$

*Proof.* The DA-PG gradient estimator is given by Equation (10):

$$\hat{\nabla}_{\boldsymbol{\theta}}^{\text{DA-PG}} \hat{J}(\bar{\pi}_{\boldsymbol{\theta}}; S_t) = \nabla_{\boldsymbol{\theta}} \bar{\pi}_{\boldsymbol{\theta}}(S_t)^{\top} \nabla_U \bar{Q}_{\mathbf{w}}(S_t, U)|_{U = \bar{\pi}_{\boldsymbol{\theta}}(S_t)}.$$

Given the conditions:

1. $\mathcal{U} = \mathcal{A}$: The distribution-parameter space is the action space.

2. $f(\cdot|u) = \delta(\cdot - u)$: Sampling $A \sim f(\cdot|u)$ yields $A = u$.

Under these conditions, $\bar{\pi}_{\boldsymbol{\theta}}(S_t)$ outputs parameters $U \in \mathcal{U}$, which are directly actions in $\mathcal{A}$. Thus, we can write $\pi_{\boldsymbol{\theta}}(S_t) = \bar{\pi}_{\boldsymbol{\theta}}(S_t)$, where $\pi_{\boldsymbol{\theta}}(S_t) \in \mathcal{A}$.

Next, consider the DA value function $\bar{q}_{\bar{\pi}_{\boldsymbol{\theta}}}(S_t, U)$. From Proposition 3.2, $\bar{q}_{\bar{\pi}_{\boldsymbol{\theta}}}(S_t, U) = \mathbb{E}_{A \sim f(\cdot|U)}\big[q_{\pi_{\boldsymbol{\theta}}}(S_t, A)\big]$. Since $f(A|U) = \delta(A - U)$, the expectation becomes $q_{\pi_{\boldsymbol{\theta}}}(S_t, U)$. So, $\bar{q}_{\bar{\pi}_{\boldsymbol{\theta}}}(S_t, U) = q_{\pi_{\boldsymbol{\theta}}}(S_t, U)$, where $U \in \mathcal{U} = \mathcal{A}$.

This means the DA critic $\bar{Q}_{\mathbf{w}}(S_t, U)$ is estimating the action-value function $q_{\pi_{\boldsymbol{\theta}}}(S_t, U)$. Thus, we can write $\bar{Q}_{\mathbf{w}}(S_t, U) = Q_{\mathbf{w}}(S_t, U)$, where $U \in \mathcal{A}$.

Substituting these equivalences into the DA-PG gradient estimator:

$$\hat{\nabla}_{\boldsymbol{\theta}}^{\text{DA-PG}} \hat{J}(\bar{\pi}_{\boldsymbol{\theta}}; S_t) = \nabla_{\boldsymbol{\theta}} \pi_{\boldsymbol{\theta}}(S_t)^{\top} \nabla_A Q_{\mathbf{w}}(S_t, A)|_{A = \pi_{\boldsymbol{\theta}}(S_t)}.$$

This is precisely the DPG gradient estimator $\hat{\nabla}_{\boldsymbol{\theta}}^{\text{DPG}} \hat{J}(\pi_{\boldsymbol{\theta}}; S_t)$ (Equation (6)). $\qquad\square$

**Proposition 4.4.** *Assume* $Q_{\mathbf{w}} = q_{\pi_{\boldsymbol{\theta}}}$ *in* $\hat{\nabla}_{\boldsymbol{\theta}}^{LR} \hat{J}(\pi_{\boldsymbol{\theta}}; S_t, A)$ *and* $\bar{Q}_{\mathbf{w}} = \bar{q}_{\bar{\pi}_{\boldsymbol{\theta}}}$ *in* $\hat{\nabla}_{\boldsymbol{\theta}}^{DA\text{-}PG} \hat{J}(\bar{\pi}_{\boldsymbol{\theta}}; S_t)$. *Then,* $\hat{\nabla}_{\boldsymbol{\theta}}^{DA\text{-}PG} \hat{J}(\bar{\pi}_{\boldsymbol{\theta}}; S_t) = \mathbb{E}_{A \sim \pi_{\boldsymbol{\theta}}(\cdot|S_t)}\big[\hat{\nabla}_{\boldsymbol{\theta}}^{LR} \hat{J}(\pi_{\boldsymbol{\theta}}; S_t, A)\big]$. *Further, if the expectation of the action-conditioned variance is greater than zero, then* $\mathbb{V}\big(\hat{\nabla}_{\boldsymbol{\theta}}^{DA\text{-}PG} \hat{J}(\bar{\pi}_{\boldsymbol{\theta}}; S_t)\big) < \mathbb{V}\big(\hat{\nabla}_{\boldsymbol{\theta}}^{LR} \hat{J}(\pi_{\boldsymbol{\theta}}; S_t, A)\big)$.

*Proof.* Proposition 3.2 states $\bar{q}_{\bar{\pi}_{\boldsymbol{\theta}}}(S_t, U) = \mathbb{E}_{A \sim f(\cdot|U)}\big[q_{\pi_{\boldsymbol{\theta}}}(S_t, A)\big]$. Given $\bar{Q}_{\mathbf{w}} = \bar{q}_{\bar{\pi}_{\boldsymbol{\theta}}}$ and $Q_{\mathbf{w}} = q_{\pi_{\boldsymbol{\theta}}}$, this becomes $\bar{Q}_{\mathbf{w}}(S_t, U) = \mathbb{E}_{A \sim f(\cdot|U)}\big[Q_{\mathbf{w}}(S_t, A)\big]$. Note that $Q_{\mathbf{w}}(S_t, A)$ and $\bar{Q}_{\mathbf{w}}(S_t, U)$ are distinct critic functions. The use of $\mathbf{w}$ for both signifies that they are learned approximators. In the context of this proof, we can think of $Q_{\mathbf{w}}$ and $\bar{Q}_{\mathbf{w}}$ as separate approximators, each utilizing a corresponding subset of $\mathbf{w}$.

Starting with the DA-PG estimator (assuming continuous $\mathcal{A}$; other cases are similar),

$$\hat{\nabla}_{\boldsymbol{\theta}}^{\text{DA-PG}} \hat{J}(\bar{\pi}_{\boldsymbol{\theta}}; S_t) = \nabla_{\boldsymbol{\theta}} \bar{\pi}_{\boldsymbol{\theta}}(S_t)^{\top} \nabla_U \bar{Q}_{\mathbf{w}}(S_t, U)|_{U = \bar{\pi}_{\boldsymbol{\theta}}(S_t)}$$

$$= \nabla_{\boldsymbol{\theta}} \bar{\pi}_{\boldsymbol{\theta}}(S_t)^{\top} \nabla_U \mathbb{E}_{A \sim f(\cdot|U)}\big[Q_{\mathbf{w}}(S_t, A)\big]\big|_{U = \bar{\pi}_{\boldsymbol{\theta}}(S_t)}$$

$$= \nabla_{\boldsymbol{\theta}} \bar{\pi}_{\boldsymbol{\theta}}(S_t)^{\top} \left(\nabla_U \int_{\mathcal{A}} f(A|U) Q_{\mathbf{w}}(S_t, A)\, dA\right)\bigg|_{U = \bar{\pi}_{\boldsymbol{\theta}}(S_t)}$$

$$= \nabla_{\boldsymbol{\theta}} \bar{\pi}_{\boldsymbol{\theta}}(S_t)^{\top} \left(\int_{\mathcal{A}} \nabla_U f(A|U) Q_{\mathbf{w}}(S_t, A)\, dA\right)\bigg|_{U = \bar{\pi}_{\boldsymbol{\theta}}(S_t)}$$

$$= \nabla_{\boldsymbol{\theta}} \bar{\pi}_{\boldsymbol{\theta}}(S_t)^{\top} \int_{\mathcal{A}} \nabla_U f(A|U)|_{U = \bar{\pi}_{\boldsymbol{\theta}}(S_t)} Q_{\mathbf{w}}(S_t, A)\, dA$$

$$= \int_{\mathcal{A}} \nabla_{\boldsymbol{\theta}} \bar{\pi}_{\boldsymbol{\theta}}(S_t)^{\top} \nabla_U f(A|U)|_{U = \bar{\pi}_{\boldsymbol{\theta}}(S_t)} Q_{\mathbf{w}}(S_t, A)\, dA$$

$$= \int_{\mathcal{A}} \nabla_{\boldsymbol{\theta}} f(A|\bar{\pi}_{\boldsymbol{\theta}}(S_t)) Q_{\mathbf{w}}(S_t, A)\, dA.$$

The differentiability under the integral sign is justified by Assumption 4.1. The last line follows from the chain rule, where $\nabla_{\boldsymbol{\theta}} f(A|\bar{\pi}_{\boldsymbol{\theta}}(S_t)) = \nabla_{\boldsymbol{\theta}} \bar{\pi}_{\boldsymbol{\theta}}(S_t)^{\top} \nabla_U f(A|U)|_{U = \bar{\pi}_{\boldsymbol{\theta}}(S_t)}$.

Using $\pi_{\boldsymbol{\theta}}(A|S_t) = f(A|\bar{\pi}_{\boldsymbol{\theta}}(S_t))$ and the log-derivative trick, we can express the DA-PG estimator as

$$\hat{\nabla}_{\boldsymbol{\theta}}^{\text{DA-PG}} \hat{J}(\bar{\pi}_{\boldsymbol{\theta}}; S_t) = \int_{\mathcal{A}} \nabla_{\boldsymbol{\theta}} \pi_{\boldsymbol{\theta}}(A|S_t) Q_{\mathbf{w}}(S_t, A)\, dA$$

$$= \int_{\mathcal{A}} \nabla_{\boldsymbol{\theta}} \log \pi_{\boldsymbol{\theta}}(A|S_t) \pi_{\boldsymbol{\theta}}(A|S_t) Q_{\mathbf{w}}(S_t, A)\, dA$$

$$= \mathbb{E}_{A \sim \pi_{\boldsymbol{\theta}}(\cdot|S_t)}\big[\nabla_{\boldsymbol{\theta}} \log \pi_{\boldsymbol{\theta}}(A|S_t) Q_{\mathbf{w}}(S_t, A)\big].$$

The LR estimator is $\hat{\nabla}_{\boldsymbol{\theta}}^{\text{LR}} \hat{J}(\pi_{\boldsymbol{\theta}}; S_t, A) = \nabla_{\boldsymbol{\theta}} \log \pi_{\boldsymbol{\theta}}(A|S_t)(Q_{\mathbf{w}}(S_t, A) - V(S_t))$. Its expectation is $\mathbb{E}_{A \sim \pi_{\boldsymbol{\theta}}(\cdot|S_t)}\big[\hat{\nabla}_{\boldsymbol{\theta}}^{\text{LR}} \hat{J}(\pi_{\boldsymbol{\theta}}; S_t, A)\big]$. The term involving the baseline $V(S_t)$ vanishes in expectation:

$$\mathbb{E}_{A \sim \pi_{\boldsymbol{\theta}}(\cdot|S_t)}\big[\nabla_{\boldsymbol{\theta}} \log \pi_{\boldsymbol{\theta}}(A|S_t) V(S_t)\big] = V(S_t) \mathbb{E}_{A \sim \pi_{\boldsymbol{\theta}}(\cdot|S_t)}\big[\nabla_{\boldsymbol{\theta}} \log \pi_{\boldsymbol{\theta}}(A|S_t)\big]$$

$$= V(S_t) \int_{\mathcal{A}} \nabla_{\boldsymbol{\theta}} \pi_{\boldsymbol{\theta}}(A|S_t)\, dA$$

$$= V(S_t) \nabla_{\boldsymbol{\theta}} \int_{\mathcal{A}} \pi_{\boldsymbol{\theta}}(A|S_t)\, dA = V(S_t) \nabla_{\boldsymbol{\theta}}(1) = 0.$$

Thus, $\mathbb{E}_{A\sim\pi_{\boldsymbol{\theta}}(\cdot|S_t)}\big[\hat{\nabla}_{\boldsymbol{\theta}}^{\text{LR}}\hat{J}(\pi_{\boldsymbol{\theta}};S_t,A)\big] = \mathbb{E}_{A\sim\pi_{\boldsymbol{\theta}}(\cdot|S_t)}\big[\nabla_{\boldsymbol{\theta}}\log\pi_{\boldsymbol{\theta}}(A|S_t)Q_{\mathbf{w}}(S_t,A)\big]$. This shows $\hat{\nabla}_{\boldsymbol{\theta}}^{\text{DA-PG}}\hat{J}(\bar{\pi}_{\boldsymbol{\theta}};S_t) = \mathbb{E}_{A\sim\pi_{\boldsymbol{\theta}}(\cdot|S_t)}\big[\hat{\nabla}_{\boldsymbol{\theta}}^{\text{LR}}\hat{J}(\pi_{\boldsymbol{\theta}};S_t,A)\big]$.

For variance reduction, let $X = \hat{\nabla}_{\boldsymbol{\theta}}^{\text{LR}}\hat{J}(\pi_{\boldsymbol{\theta}};S_t,A)$ and $Y = \hat{\nabla}_{\boldsymbol{\theta}}^{\text{DA-PG}}\hat{J}(\bar{\pi}_{\boldsymbol{\theta}};S_t)$. We have $Y = \mathbb{E}\big[X|S_t,\bar{\pi}_{\boldsymbol{\theta}}(S_t)\big]$ (expectation over $A$). By the law of total variance: $\mathbb{V}\big(X\big) = \mathbb{E}\big[\mathbb{V}\big(X|S_t,\bar{\pi}_{\boldsymbol{\theta}}(S_t)\big)\big] + \mathbb{V}\big(\mathbb{E}\big[X|S_t,\bar{\pi}_{\boldsymbol{\theta}}(S_t)\big]\big)$. This translates to

$$\mathbb{V}\big(\hat{\nabla}_{\boldsymbol{\theta}}^{\text{LR}}\hat{J}(\pi_{\boldsymbol{\theta}};S_t,A)\big) = \mathbb{E}_{S_t}\big[\mathbb{V}_A\big(\hat{\nabla}_{\boldsymbol{\theta}}^{\text{LR}}\hat{J}(\pi_{\boldsymbol{\theta}};S_t,A)|S_t\big)\big] + \mathbb{V}\big(\hat{\nabla}_{\boldsymbol{\theta}}^{\text{DA-PG}}\hat{J}(\bar{\pi}_{\boldsymbol{\theta}};S_t)\big).$$

If $\mathbb{E}_{S_t}\big[\mathbb{V}_A\big(\hat{\nabla}_{\boldsymbol{\theta}}^{\text{LR}}\hat{J}(\pi_{\boldsymbol{\theta}};S_t,A)|S_t\big)\big] > 0$ (i.e., the action-conditioned variance is positive on average), then $\mathbb{V}\big(\hat{\nabla}_{\boldsymbol{\theta}}^{\text{DA-PG}}\hat{J}(\bar{\pi}_{\boldsymbol{\theta}};S_t)\big) < \mathbb{V}\big(\hat{\nabla}_{\boldsymbol{\theta}}^{\text{LR}}\hat{J}(\pi_{\boldsymbol{\theta}};S_t,A)\big)$. $\qquad\square$

**Proposition 4.5.** *Assume $\mathcal{A}$ is continuous, $Q_{\mathbf{w}} = q_{\pi_{\boldsymbol{\theta}}}$ in $\hat{\nabla}_{\boldsymbol{\theta}}^{RP}\hat{J}(\pi_{\boldsymbol{\theta}};S_t,\epsilon)$, and $\bar{Q}_{\mathbf{w}} = \bar{q}_{\bar{\pi}_{\boldsymbol{\theta}}}$ in $\hat{\nabla}_{\boldsymbol{\theta}}^{DA\text{-}PG}\hat{J}(\bar{\pi}_{\boldsymbol{\theta}};S_t)$. Then, $\hat{\nabla}_{\boldsymbol{\theta}}^{DA\text{-}PG}\hat{J}(\bar{\pi}_{\boldsymbol{\theta}};S_t) = \mathbb{E}_{\epsilon\sim p}\big[\hat{\nabla}_{\boldsymbol{\theta}}^{RP}\hat{J}(\pi_{\boldsymbol{\theta}};S_t,\epsilon)\big]$. Further, if the expectation of the noise-induced variance is greater than zero, then $\mathbb{V}\big(\hat{\nabla}_{\boldsymbol{\theta}}^{DA\text{-}PG}\hat{J}(\bar{\pi}_{\boldsymbol{\theta}};S_t)\big) < \mathbb{V}\big(\hat{\nabla}_{\boldsymbol{\theta}}^{RP}\hat{J}(\pi_{\boldsymbol{\theta}};S_t,\epsilon)\big)$.*

*Proof.* For the RP estimator, the action is generated as $A = g_{\boldsymbol{\theta}}(\epsilon;S_t)$, where $\epsilon \sim p(\cdot)$. For consistency with DA-PG notation, we can write $A = g(\epsilon;U)$, where $U = \bar{\pi}_{\boldsymbol{\theta}}(S_t) \in \mathcal{U}$ represents all relevant learnable distribution parameters. Thus, the distribution $f(\cdot|U)$ of the random variable $A$ is induced by $g(\epsilon;U)$ with $\epsilon \sim p(\cdot)$.

Similar to the proof of Proposition 4.4, given the critics are the corresponding true action-value functions, we have

$$\bar{Q}_{\mathbf{w}}(S_t,U) = \mathbb{E}_{A\sim f(\cdot|U)}\big[Q_{\mathbf{w}}(S_t,A)\big] = \mathbb{E}_{\epsilon\sim p}\big[Q_{\mathbf{w}}(S_t,g(\epsilon;\bar{\pi}_{\boldsymbol{\theta}}(S_t)))\big],$$

where we use a change of variables to express the expectation in terms of the noise $\epsilon$.

Now, we can express the DA-PG gradient as

$$\begin{aligned}
\hat{\nabla}_{\boldsymbol{\theta}}^{\text{DA-PG}}\hat{J}(\bar{\pi}_{\boldsymbol{\theta}};S_t) &= \nabla_{\boldsymbol{\theta}}\bar{\pi}_{\boldsymbol{\theta}}(S_t)^{\top}\nabla_U\bar{Q}_{\mathbf{w}}(S_t,U)|_{U=\bar{\pi}_{\boldsymbol{\theta}}(S_t)} \\
&= \nabla_{\boldsymbol{\theta}}\bar{\pi}_{\boldsymbol{\theta}}(S_t)^{\top}\nabla_U\mathbb{E}_{\epsilon\sim p}\big[Q_{\mathbf{w}}(S_t,g(\epsilon;U))\big]\big|_{U=\bar{\pi}_{\boldsymbol{\theta}}(S_t)} \\
&= \nabla_{\boldsymbol{\theta}}\bar{\pi}_{\boldsymbol{\theta}}(S_t)^{\top}\mathbb{E}_{\epsilon\sim p}\big[\nabla_U Q_{\mathbf{w}}(S_t,g(\epsilon;U))|_{U=\bar{\pi}_{\boldsymbol{\theta}}(S_t)}\big] \\
&= \mathbb{E}_{\epsilon\sim p}\big[\nabla_{\boldsymbol{\theta}}\bar{\pi}_{\boldsymbol{\theta}}(S_t)^{\top}\nabla_U Q_{\mathbf{w}}(S_t,g(\epsilon;U))|_{U=\bar{\pi}_{\boldsymbol{\theta}}(S_t)}\big] \\
&= \mathbb{E}_{\epsilon\sim p}\big[\nabla_{\boldsymbol{\theta}}Q_{\mathbf{w}}(S_t,g(\epsilon;\bar{\pi}_{\boldsymbol{\theta}}(S_t)))\big].
\end{aligned}$$

The differentiability under the integral sign is justified by Assumption 4.1. The last line follows from the chain rule, where $\nabla_{\boldsymbol{\theta}}Q_{\mathbf{w}}(S_t,g(\epsilon;\bar{\pi}_{\boldsymbol{\theta}}(S_t))) = \nabla_{\boldsymbol{\theta}}\bar{\pi}_{\boldsymbol{\theta}}(S_t)^{\top}\nabla_U Q_{\mathbf{w}}(S_t,g(\epsilon;U))|_{U=\bar{\pi}_{\boldsymbol{\theta}}(S_t)}$.

On the other hand, the RP gradient is

$$\begin{aligned}
\hat{\nabla}_{\boldsymbol{\theta}}^{\text{RP}}\hat{J}(\pi_{\boldsymbol{\theta}};S_t,\epsilon) &= \nabla_{\boldsymbol{\theta}}g_{\boldsymbol{\theta}}(\epsilon;S_t)^{\top}\nabla_A Q_{\mathbf{w}}(S_t,A)|_{A=g_{\boldsymbol{\theta}}(\epsilon;S_t)} \\
&= \nabla_{\boldsymbol{\theta}}g(\epsilon;\bar{\pi}_{\boldsymbol{\theta}}(S_t))^{\top}\nabla_A Q_{\mathbf{w}}(S_t,A)|_{A=g(\epsilon;\bar{\pi}_{\boldsymbol{\theta}}(S_t))} \\
&= \nabla_{\boldsymbol{\theta}}Q_{\mathbf{w}}(S_t,g(\epsilon;\bar{\pi}_{\boldsymbol{\theta}}(S_t))),
\end{aligned}$$

where we use the chain rule again in the last equation: $\nabla_{\boldsymbol{\theta}}Q_{\mathbf{w}}(S_t,g(\epsilon;\bar{\pi}_{\boldsymbol{\theta}}(S_t))) = \nabla_{\boldsymbol{\theta}}g(\epsilon;\bar{\pi}_{\boldsymbol{\theta}}(S_t))^{\top}\nabla_A Q_{\mathbf{w}}(S_t,A)|_{A=g(\epsilon;\bar{\pi}_{\boldsymbol{\theta}}(S_t))}$. Thus, we have

$$\hat{\nabla}_{\boldsymbol{\theta}}^{\text{DA-PG}}\hat{J}(\bar{\pi}_{\boldsymbol{\theta}};S_t) = \mathbb{E}_{\epsilon\sim p}\big[\hat{\nabla}_{\boldsymbol{\theta}}^{\text{RP}}\hat{J}(\pi_{\boldsymbol{\theta}};S_t,\epsilon)\big].$$

The variance reduction argument is similar to that in Proposition 4.4. Let $X = \hat{\nabla}_{\boldsymbol{\theta}}^{\text{RP}}\hat{J}(\pi_{\boldsymbol{\theta}};S_t,\epsilon)$ and $Y = \hat{\nabla}_{\boldsymbol{\theta}}^{\text{DA-PG}}\hat{J}(\bar{\pi}_{\boldsymbol{\theta}};S_t)$. We have $Y = \mathbb{E}\big[X|S_t,\epsilon\big]$ (expectation over $\epsilon$). By the law of total variance: $\mathbb{V}\big(X\big) = \mathbb{E}\big[\mathbb{V}\big(X|S_t,\epsilon\big)\big] + \mathbb{V}\big(\mathbb{E}\big[X|S_t,\epsilon\big]\big)$. This translates to

$$\mathbb{V}\big(\hat{\nabla}_{\boldsymbol{\theta}}^{\text{RP}}\hat{J}(\pi_{\boldsymbol{\theta}};S_t,\epsilon)\big) = \mathbb{E}_{S_t}\big[\mathbb{V}_{\epsilon}\big(\hat{\nabla}_{\boldsymbol{\theta}}^{\text{RP}}\hat{J}(\pi_{\boldsymbol{\theta}};S_t,\epsilon)|S_t\big)\big] + \mathbb{V}\big(\hat{\nabla}_{\boldsymbol{\theta}}^{\text{DA-PG}}\hat{J}(\bar{\pi}_{\boldsymbol{\theta}};S_t)\big).$$

If $\mathbb{E}_{S_t}\big[\mathbb{V}_{\epsilon}\big(\hat{\nabla}_{\boldsymbol{\theta}}^{\text{RP}}\hat{J}(\pi_{\boldsymbol{\theta}};S_t,\epsilon)|S_t\big)\big] > 0$ (i.e., the noise-induced variance is positive on average), then $\mathbb{V}\big(\hat{\nabla}_{\boldsymbol{\theta}}^{\text{DA-PG}}\hat{J}(\bar{\pi}_{\boldsymbol{\theta}};S_t)\big) < \mathbb{V}\big(\hat{\nabla}_{\boldsymbol{\theta}}^{\text{RP}}\hat{J}(\pi_{\boldsymbol{\theta}};S_t,\epsilon)\big)$. $\qquad\square$

## C.3 CONVERGENCE ANALYSIS FOR DA-PG

We present a convergence result for the distributions-as-actions policy gradient (DA-PG), which is a direct application of the convergence of the deterministic policy gradient (DPG; Xiong et al., 2022). We assume an on-policy linear function approximation setting and use TD learning to learn the critic. See Algorithm 1 for the analyzed DA-PG-TD algorithm. We follow the notation of Xiong et al. as much as possible for comparison with their results. As an example, instead of using $d_{\bar{\pi}_\theta}$ to represent the discounted occupancy measure under $\bar{\pi}_\theta$ as in other parts of this paper, we use the notation of Xiong et al. and refer to it as $\nu_\theta$ in this section.

---

**Algorithm 1** DA-PG-TD

---

1: **Input:** $\alpha_w, \alpha_\theta, w_0, \theta_0$, batch size $M$.
2: **for** $t = 0, 1, \ldots, T - 1$ **do**
3:     **for** $j = 0, 1, \ldots, M - 1$ **do**
4:         Sample $s_{t,j} \sim d_{\theta_t}$.
5:         Generate $u_{t,j} = \bar{\pi}_{\theta_t}(s_{t,j})$.
6:         Sample $s_{t+1,j} \sim \bar{p}(\cdot|s_{t,j}, u_{t,j})$ and $r_{t,j}$.
7:         Generate $u_{t+1,j} = \bar{\pi}_{\theta_t}(s_{t+1,j})$.
8:         Denote $x_{t,j} = (s_{t,j}, u_{t,j})$.
9:         $\delta_{t,j} = r_{t,j} + \gamma \phi(x_{t+1,j})^\top w_t - \phi(x_{t,j})^\top w_t$.
10:     **end for**
11:     $w_{t+1} = w_t + \frac{\alpha_w}{M} \sum_{j=0}^{M-1} \delta_{t,j} \phi(x_{t,j})$.
12:     **for** $j = 0, 1, \ldots, M - 1$ **do**
13:         Sample $s'_{t,j} \sim \nu_{\theta_t}$.
14:     **end for**
15:     $\theta_{t+1} = \theta_t + \frac{\alpha_\theta}{M} \sum_{j=0}^{M-1} \nabla_\theta \bar{\pi}_{\theta_t}(s'_{t,j}) \nabla_\theta \bar{\pi}_{\theta_t}(s'_{t,j})^\top w_t$.
16: **end for**

---

Following their notation, the parameterized policy is denoted as $\bar{\pi}_\theta$ and the objective function $J(\bar{\pi}_\theta)$ (Equation (1)) is denoted as $J(\theta)$. The distributions-as-actions policy gradient is

$$\nabla_\theta J(\theta) = \mathbb{E}_{s \sim \nu_\theta} \left[ \nabla_\theta \bar{\pi}_\theta(s) \nabla_u \bar{q}_{\bar{\pi}_\theta}(s, u)|_{u = \bar{\pi}_\theta(s)} \right], \tag{17}$$

where $\nu_\theta(s) \doteq \sum_{t=0}^{\infty} \mathbb{E}_{\bar{\pi}_\theta} \left[ \gamma^t \mathbb{I}(S_t = s) \right]$ is the discounted occupancy measure under $\bar{\pi}_\theta$. We also define the stationary distribution of $\bar{\pi}_\theta$ to be $d_\theta(s) = \lim_{T \to \infty} \frac{1}{T} \sum_{t=0}^{T-1} \mathbb{E}_{\bar{\pi}_\theta} \left[ \mathbb{I}(S_t = s) \right]$. Under linear function approximation for the critic function, the parameterized critic can be expressed as $\bar{Q}_w(s, u) = \phi(s, u)^\top w$, where $\phi : \mathcal{S} \times \mathcal{U} \to \mathbb{R}^d$ is the feature function.

We will first list the full set of assumptions needed for the convergence result, followed by the convergence theorem. In addition, we incorporate the corrections to the result of Xiong et al. from Vasan et al. (2024), which extends the result to the reparameterization policy gradient. Following Vasan et al., the corrections are highlighted in red.

**Assumption C.1.** For any $\theta_1, \theta_2, \theta \in \mathbb{R}^d$, there exist positive constants $L_{\bar{\pi}}, L_\phi$ and $\lambda_\Phi$, such that (1) $\|\bar{\pi}_{\theta_1}(s) - \bar{\pi}_{\theta_2}(s)\| \leq L_{\bar{\pi}} \|\theta_1 - \theta_2\|, \forall s \in \mathcal{S}$; (2) $\|\nabla_\theta \bar{\pi}_{\theta_1}(s) - \nabla_\theta \bar{\pi}_{\theta_2}(s)\| \leq L_\psi \|\theta_1 - \theta_2\|, \forall s \in \mathcal{S}$; (3) the matrix $\Psi_\theta := \mathbb{E}_{\nu_\theta} \left[ \nabla_\theta \bar{\pi}_\theta(s) \nabla_\theta \bar{\pi}_\theta(s)^\top \right]$ is non-singular with the minimal eigenvalue uniformly lower-bounded as $\sigma_{\min}(\Psi_\theta) \geq \lambda_\Psi$.

**Assumption C.2.** For any $u_1, u_2 \in \mathcal{U}$, there exist positive constants $L_{\bar{p}}, L_{\bar{r}}$, such that (1) the distributions-as-actions transition kernel satisfies $|\bar{p}(s'|s, u_1) - \bar{p}(s'|s, u_2)| \leq L_{\bar{p}} \|u_1 - u_2\|, \forall s, s' \in \mathcal{S}$; (2) the distributions-as-actions reward function satisfies $|\bar{r}(s, u_1) - \bar{r}(s, u_2)| \leq L_{\bar{r}} \|u_1 - u_2\|, \forall s, s' \in \mathcal{S}$.

**Assumption C.3.** For any $u_1, u_2 \in \mathcal{U}$, there exists a positive constant $L_{\bar{q}}$, such that $\|\nabla_u \bar{q}_{\bar{\pi}_\theta}(s, u_1) - \nabla_u \bar{q}_{\bar{\pi}_\theta}(s, u_2)\| \leq L_{\bar{q}} \|u_1 - u_2\|, \forall \theta \in \mathbb{R}^d, s \in \mathcal{S}$.

**Assumption C.4.** The feature function $\phi : \mathcal{S} \times \mathcal{U} \to \mathbb{R}^d$ is uniformly bounded, i.e., $\|\phi(\cdot, \cdot)\| \leq C_\phi$ for some positive constant $C_\phi$. In addition, we define $A = \mathbb{E}_{d_\theta} \left[ \phi(x)(\gamma \phi(x') - \phi(x))^\top \right]$ and $D = \mathbb{E}_{d_\theta} \left[ \phi(x)\phi(x)^\top \right]$, and assume that $A$ and $D$ are non-singular. We further assume that the absolute value of the eigenvalues of $A$ are uniformly lower bounded, i.e., $|\sigma(A)| \geq \lambda_A$ for some positive constant $\lambda_A$.

**Proposition C.5** (Compatible function approximation). *A function estimator $\bar{Q}_w(s,u)$ is compatible with a policy $\bar{\pi}_\theta$, i.e., $\nabla J(\theta) = \mathbb{E}_{\nu_\theta}\left[\nabla_\theta \bar{\pi}_\theta(s)\nabla_u \bar{Q}_w(s,u)|_{u=\bar{\pi}_\theta(s)}\right]$, if it satisfies the following two conditions:*

1. *$\nabla_u \bar{Q}_w(s,u)|_{u=\bar{\pi}_\theta(s)} = \nabla_\theta \bar{\pi}_\theta(s)^\top w$;*

2. *$w = w^*_{\xi_\theta}$ minimizes the mean square error $\mathbb{E}_{\nu_\theta}\left[\xi(s;\theta,w)^\top \xi(s;\theta,w)\right]$, where $\xi(s;\theta,w) = \nabla_u \bar{Q}_w(s,u)|_{u=\bar{\pi}_\theta(s)} - \nabla_u \bar{q}_{\bar{\pi}_\theta}(s,u)|_{u=\bar{\pi}_\theta(s)}$.*

Given the above assumption, one can show that the distributions-as-actions policy gradient is smooth (Lemma C.6), and that Algorithm 1 converges (Theorem C.7).

**Lemma C.6.** *Suppose Assumptions C.1-C.3 hold. Then the distributions-as-actions policy gradient $\nabla J(\theta)$ defined in Equation (17) is Lipschitz continuous with the parameter $L_J$, i.e., $\forall \theta_1, \theta_2 \in \mathbb{R}^d$,*

$$\|\nabla J(\theta_1) - \nabla J(\theta_2)\| \le L_J \|\theta_1 - \theta_2\|,$$

*where $L_J = \left(\frac{1}{2}L_{\bar{p}}L_{\bar{\pi}}^2 L_\nu C_\nu + \frac{L_\psi}{1-\gamma}\right)\left(L_{\bar{r}} + \frac{\gamma R_{\max}L_{\bar{p}}}{1-\gamma}\right) + \frac{L_{\bar{\pi}}}{1-\gamma}\left(L_{\bar{q}}L_{\bar{\pi}} + \frac{\gamma}{2}L_{\bar{p}}^2 R_{\max}L_{\bar{\pi}}C_\nu + \frac{\gamma L_{\bar{p}}L_{\bar{r}}L_{\bar{\pi}}}{1-\gamma}\right)$.*

**Theorem C.7.** *Suppose that Assumptions C.1-C.4 hold. Let $\alpha_w \le \frac{\lambda}{2C_A^2}$; $M \ge \frac{48\alpha_w C_A^2}{\lambda}$; $\alpha_\theta \le$ $\min\left\{\frac{1}{4L_J}, \frac{\lambda\alpha_w}{24\sqrt{6}L_h L_w}\right\}$. Then the output of DA-PG-TD in Algorithm 1 satisfies*

$$\min_{t \in [T]} \mathbb{E}\|\nabla J(\theta_t)\|^2 \le \frac{c_1}{T} + \frac{c_2}{M} + c_3 \kappa^2,$$

*where $c_1 = \frac{8R_{\max}}{\alpha_\theta(1-\gamma)} + \frac{144L_h^2}{\lambda\alpha_w}\|w_0 - w^*_{\theta_0}\|^2$, $c_2 = \left[48\alpha_w^2(C_A^2 C_w^2 + C_b^2) + \frac{96L_w^2 L_{\bar{\pi}}^4 C_{w_\xi}^2 \alpha_\theta^2}{\lambda\alpha_w}\right] \cdot \frac{144L_h^2}{\lambda\alpha_w} + 72L_{\bar{\pi}}^4 C_{w_\xi}^2$, $c_3 = 18L_h^2 + \left[\frac{24L_w^2 L_h^2 \alpha_\theta^2}{\lambda\alpha_w} + \frac{24}{\lambda\alpha_w}\right] \cdot \frac{144L_h^2}{\lambda\alpha_w}$ with $C_A = 2C_\phi^2, C_b = R_{\max}C_\phi, C_w = \frac{R_{\max}C_\phi}{\lambda_A}, C_{w_\xi} = \frac{L_{\bar{\pi}}C_{\bar{q}}}{\lambda_\Psi(1-\gamma)}, L_w = \frac{L_J}{\lambda_\Psi} + \frac{L_{\bar{\pi}}C_{\bar{q}}}{\lambda_\Psi^2(1-\gamma)}\left(L_{\bar{\pi}}^2 L_\nu + \frac{2L_{\bar{\pi}}L_\psi}{1-\gamma}\right), L_h = L_{\bar{\pi}}^2, C_{\bar{q}} = L_{\bar{r}} + L_{\bar{p}} \cdot \frac{\gamma R_{\max}}{1-\gamma}, L_\nu = \frac{1}{2}C_\nu L_{\bar{p}}L_{\bar{\pi}}$, and $L_J$ defined in Lemma C.6, and we define*

$$\kappa := \max_\theta \|w^*_\theta - w^*_{\xi_\theta}\|.$$

*Remark* C.8. Apart from the corrections highlighted in red, the convergence result retains the same mathematical form as the DPG convergence result (see Theorem 1 of Xiong et al. (2022)). However, the associated constants differ, as they are defined with respect to the distributions-as-actions formulations of the MDP, policy, and critic. Notably, the distributions-as-actions policy class strictly generalizes the deterministic policy class. Consequently, this convergence result constitutes a strict generalization of the DPG convergence result.

The proofs of Lemma C.6 and Theorem C.7 follow the same lines as that of Lemma 1 and Theorem 1 of Xiong et al. (2022). We refer the reader to Xiong et al. for proofs and discussion and Vasan et al. (2024) for details about the corrections.

## D EXPERIMENTAL DETAILS

Our implementation builds upon a PyTorch (Paszke, 2019) implementation of TD3 from CleanRL (Huang et al., 2022), distributed under the MIT license.

Since the performance distribution in reinforcement learning (RL) is often not Gaussian, we use $95\%$ bootstrap confidence intervals (CIs) for reporting the statistical significance whenever applicable, as recommended by Patterson et al. (2024). We use scipy.stats.bootstrap with $10,000$ resamples from SciPy (Virtanen et al., 2020) to calculate the bootstrap CIs.

For all bar plots, we plot the final performance, which is computed using the average of the return collected during the final $10\%$ training steps. We treat each seed in each environment as an independent trial and compute bootstrap CIs over the variability across both seeds and environments, thereby capturing both stochastic training variation and environment-specific effects.

## D.1 POLICY PARAMETERIZATION AND ACTION SAMPLING

When the action space is multidimensional, we treat each dimension independently. For simplicity, our exposition will focus on the unidimensional case in the remaining of the paper.

**Discrete action spaces** We use the categorical policy parameterization: $A \sim f(\cdot|[p_1, \cdots, p_N]^\top)$, where $f(x|[p_1, \cdots, p_N]^\top) = \prod_{i=1}^{N} p_i^{\mathbb{I}(x=i)}$ is the probability mass function for the categorical distribution. For DA-AC, we choose the probability vector $u = [p_1, \cdots, p_N]^\top$ as the distribution parameters. We define the distribution parameters corresponding to an action $A$ to be the one-hot vector $U_A = \text{one\_hot}(A)$.

**Continuous action spaces** Assume the action space is $[a_{\min}, a_{\max}]$. We use the Gaussian policy parameterization that is used in TD3: $A = \text{clip}(\mu + \epsilon, a_{\min}, a_{\max}), \epsilon \sim \mathcal{N}(0, \sigma)$. Same as TD3, we restrict the mean $\mu$ to be within $[a_{\min}, a_{\max}]$ using a squashing function:

$$\mu = \frac{u_\mu + 1}{2}(a_{\max} - a_{\min}) + a_{\min}, \quad u_\mu = \tanh(\text{logit}_\mu), \tag{18}$$

where $\text{logit}_\mu \in \mathbb{R}$ is the actor network's output for $\mu$. While TD3 uses a fixed $\sigma_{\text{TD3}} = 0.1 * \frac{a_{\max} - a_{\min}}{2}$, we allow the learnable standard deviation to be within a range $\sigma \in [\sigma_{\min}, \sigma_{\max}]$:

$$\log \sigma = \frac{u_\sigma + 1}{2} * (\log \sigma_{\max} - \log \sigma_{\min}) + \log \sigma_{\min}, \quad u_\sigma = \tanh(\text{logit}_\sigma), \tag{19}$$

where $\text{logit}_\sigma \in \mathbb{R}$ is the actor network's output for $\sigma$. For RP-AC, the reparameterization function is $g_{\boldsymbol{\theta}}(\epsilon; S_t) = \text{clip}(\mu_{\boldsymbol{\theta}}(S_t) + \sigma_{\boldsymbol{\theta}}(S_t)\epsilon, a_{\min}, a_{\max}), \epsilon \sim \mathcal{N}(0, 1)$. For DA-AC, we choose the distribution parameters to be $u = [u_\mu, u_\sigma]^\top \in [-1, 1]^2$ so that the parameter space is consistent across the mean and standard deviation dimensions. Since we lower bound the standard deviation space to encourage exploration, we define the distribution parameters corresponding to an action $A$ to be $U_A = [\frac{2A}{a_{\max} - a_{\min}}, -1]^\top$ to approximate the Dirac delta distribution, which corresponds to $\mu = A$ and $\sigma = \sigma_{\min}$.

**Hybrid action spaces** For environments with hybrid action spaces, DA-AC simply uses the policy parameterizations described above for the corresponding discrete and continuous parts.

## D.2 POLICY EVALUATION IN BANDITS

**K-Armed Bandit** We use a K-armed bandit with $K = 3$ and a deterministic reward function:

$$r(a_1) = 0, \quad r(a_2) = 0.5, \quad r(a_3) = 1.$$

**Bimodal Continuous Bandit** We use a continuous bandit with a bimodal reward function that is deterministic. The reward function is the normalized summation of two Gaussians' density functions whose standard deviations are both $0.5$ and whose means are $-1$ and $1$, respectively:

$$r(a) = e^{-\frac{(a+1)^2}{0.5}} + e^{-\frac{(a-1)^2}{0.5}}.$$

We restrict the action space to be $[a_{\min}, a_{\max}] = [-2, 2]$. The standard standard deviation is constrained to $[\sigma_{\min}, \sigma_{\max}] = [e^{-3}, e]$, corresponding to

$$[\log \sigma_{\min}, \log \sigma_{\max}] = [-3, 1]. \tag{20}$$

**Critic network architecture** To be consistent with the RL settings, we use the same critic network architecture as those in Appendices D.4 and D.6. Specifically, we use a two-layer MLP network with the concatenated state and action vector as input. We reduce the hidden size from 256 to 16 and use a dummy state vector with a value of 1.

**Experimental details** We keep the policy evaluation (PE) policy fixed and update the distributions-as-actions critic function for 2000 steps using either Equation (11) or Equation (13). In K-Armed Bandit, the PE policy is $\bar{\pi}_{\text{PE}} = u_{\text{PE}} = [1/3, 1/3, 1/3]$; in Bimodal Continuous Bandit, the PE policy is $\bar{\pi}_{\text{PE}} = u_{\text{PE}} = [0, 0.5]$ (corresponding to $\mu = 0$ and $\log \sigma = 0.0$; see Equations (19) and (20) for the calculation of $\log \sigma$). The hyperparameters are the same as those of DA-AC in Table 4, except that the actor is kept fixed to the corresponding PE policy.

### D.3 POLICY OPTIMIZATION IN BANDITS

**Environments**    We use the same bandit environments as Appendix D.2.

**Algorithms**    In addition to DA-AC and DA-AC w/o ICL, we also include LR-AC and RP-AC as a reference, as they should be quite effective in these settings because of a much simpler critic function. Note that our goal is not to show that DA-AC can outperform other baselines in these toy settings, but rather to illustrate how ICL substantially improves critic learning in DA-AC. Here, LR-AC uses the average of the action values as the baseline. We also include LR-PG, RP-PG, and DA-PG, variants of LR-AC, RP-AC, and DA-AC that have access to their corresponding true value functions to remove the confounding factor of learning the critic.

**Experimental Details**    We use the same critic network architecture as in Appendix D.2. We use the same actor network architecture as those in Appendices D.4 and D.6. Specifically, we use a two-layer MLP network with the state vector as input. We reduce the hidden size from 256 to 16 and use a dummy state tensor with a value of 1. The hyperparameters are in Table 4. For LR-PG, RP-PG, and DA-PG, the critic function is calculated analytically; otherwise, their hyperparameters are the same as their counterparts with a learned critic function. See Figure 9 for learning curves.

**Results with alternative learning rates**    While we choose a fixed learning rate for all algorithms for a more controlled comparison in Section 5.5, we note that interpolated critic learning (ICL) also improves the performance of DA-AC under other learning rates (see Figure 10).

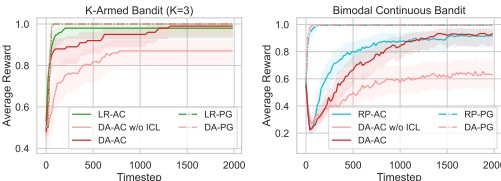

Figure 9: **Learning curves of DA-AC, DA-AC w/o ICL, and baselines** on the K-Armed Bandit (col 1) and Bimodal Continuous Bandit (col 2) tasks. Results are averaged over 50 seeds. Shaded regions show 95% bootstrap CIs. DA-PG exhibits slightly faster convergence than RP-PG—albeit marginal—highlighting the advantage of using a lower-variance estimator when no bias is present. When the critic is learned, DA-AC w/o ICL performs significantly worse than LR-AC and RP-AC, achieving highly suboptimal returns even by the end of training. This reflects the difficulty of learning an effective critic using the standard update, as discussed in Section 4.2. In contrast, although DA-AC initially learns more slowly than LR-AC and RP-AC, it substantially outperforms DA-AC w/o ICL and eventually matches LR-AC and RP-AC in the later stages of training.

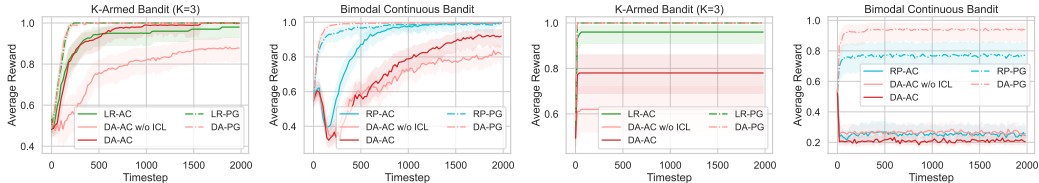

Figure 10: **Learning curves of DA-AC, DA-AC w/o ICL, and baselines** using learning rates 0.001 (cols 1–2) and 0.1 (cols 3–4). Results are averaged over 50 seeds. Shaded regions show 95% bootstrap CIs. An aggressive learning rate of 0.1 often leads to premature convergence to suboptimal points for most algorithms. Consistent with Figure 9, ICL demonstrates improved performance for DA-AC when a more conservative learning rate is employed.

### D.4 CONTINUOUS CONTROL

**Environments**    From OpenAI Gym MuJoCo, we use the most commonly used 5 environments (see Table 2). From DeepMind Control Suite, we use the same 15 environments as D'Oro et al.

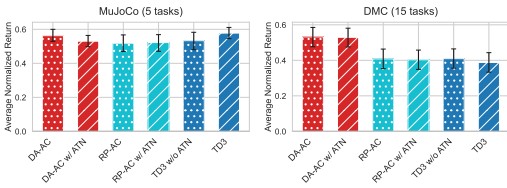

Figure 11: **Average normalized returns with and without actor target network (ATN)** on MuJoCo (col 1) and DMC (col 2) tasks. Values are averaged over 10 seeds and 5 (MuJoCo) or 10 (DMC) tasks. Error bars show 95% bootstrap CIs.

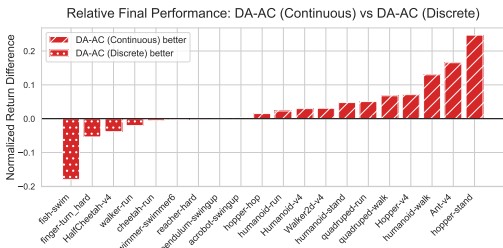

Figure 12: **Relative final performance of DA-AC with continuous actions versus with discrete actions** across 20 individual MuJoCo and DMC tasks. Results are averaged over 10 seeds per task.

(2023), which are mentioned to be neither immediately solvable nor unsolvable by common deep RL algorithms. The full list of environments and their corresponding observation and action space dimensions are in Table 3. Returns for bar plots are normalized by dividing the episodic return by the maximum possible return for a given task. In DMC environments, the maximum return is 1000 (Tunyasuvunakool et al., 2020). For MuJoCo environments, we establish maximum returns based on the highest values observed from proficient RL algorithms (Bhatt et al., 2024): 4000 for Hopper-v4, 7000 for Walker2d-v4, 8000 for Ant-v4, 16000 for HalfCheetah-v4, and 12000 for Humanoid-v4.

**Experimental details** Similar to TD3, DA-AC and RP-AC also adopt a uniform exploration phase. During the uniform exploration phase, the distribution parameters $u = [u_\mu, u_\sigma]^\top$ are uniformly sampled from $[-1, 1]^2$. These three algorithms use the default hyperparameters of TD3 (see Table 5). For SAC (Haarnoja et al., 2018) and PPO (Schulman et al., 2017), we use the implementations and tuned hyperparameters in CleanRL (Huang et al., 2022) (see Tables 5 and 8, respectively). Learning curves in each individual environment can be found in Figure 13.

**Impact of the actor target network** We also investigate the impact of using an actor target network (ATN) in DA-AC and the baselines. While TD3 already employs an ATN, both DA-AC and RP-AC do not. We additionally test DA-AC w/ ATN and RP-AC w/ ATN and TD3 w/o ATN. From Figure 11, we can see that the actor target network does not have a significant impact in general.

### D.5 DISCRETE CONTROL

**Environments** We use the same 4 Gym classic control (Brockman et al., 2016) and 5 MinAtar (Young & Tian, 2019) environments as in Ceron & Castro (2021).

**Experimental details for Gym environments** We use the existing implementations and tuned hyperparameters of DQN (Mnih et al., 2015) and PPO in CleanRL (see Tables 9 and 11, respectively). For DA-AC, ST-AC, LR-AC, and EAC, we adjust relevant off-policy training hyperparameters based on those of DQN, including batch size, gradient steps per step, network size, replay buffer size. We also disable double Q-networks to better align with DQN. See Table 6 for the updated parameters from Table 5. We use a similar setup for Discrete SAC (DSAC; Christodoulou, 2019), as shown in the same table. Learning curves can be found in Figure 14.

**Experimental details for MinAtar environments**     The MinAtar setups for DQN and PPO are adopted from their implementations and tuned hyperparameters for Atari (Bellemare et al., 2013) in CleanRL (see Tables 10 and 12, respectively). Similar to the above, we adjust relevant off-policy training hyperparameters based on DQN for DA-AC, ST-AC, LR-AC, and EAC (see Table 7). We use a similar setup for DSAC but decrease its uniform exploration steps according to its hyperparameters for Atari in CleanRL (see Table 7). For consistency, we use the same critic network for DA-AC, ST-AC, LR-AC, EAC, and DSAC, which takes actions as input. See the next paragraph for the joint encoding of CNN observation features and actions for these methods in this experiment. Learning curves can be found in Figure 14.

**Joint encoding of CNN observation features and actions**     While concatenation is used for joint encoding of observations and actions for state-based observations in MuJoCo/DMC/Gym environments, it might not be efficient for encoding latent features and actions (Schlegel et al., 2023). Inspired by Schlegel et al., we use the flattened outer product of the CNN observation features (with a dimension of 128) and the vectorized action representations (with a dimension of $|\mathcal{A}|$) as the joint encoding. The action representations are the action probabilities for DA-AC, while they are one-hot embedding of actions for other algorithms. We then use an additional hidden layer with a small number of hidden units (8 in MinAtar) with negligible overhead to extract higher-level features.

### D.6   HIGH-DIMENSIONAL DISCRETE CONTROL

**Details**     We use the same 20 environments as Appendix D.4. Similar to the continuous control case, we also include a uniform exploration phase for all discrete control algorithms. For LR-AC and ST-AC, the action is randomly sampled from a uniform categorical distribution. For DA-AC, the logits of the distribution parameters (in this case, the probability vector) are sampled from $\mathcal{N}(0, 1)^N$, where $N$ is the number of possible discrete outcomes. All algorithms use the default hyperparameters of TD3 (see Table 5). See Figure 15 for learning curves in each individual environment.

**Comparison to continuous control**     We plot the relative final performance of DA-AC with continuous actions versus with discrete actions in Figure 12. We can see that the performance of DA-AC with discrete actions can often compete with DA-AC with continuous actions.

### D.7   HYBRID CONTROL

**Environments**     We use 7 parameterized-action MDP (PAMDP; Masson et al., 2016) environments from Li et al. (2022). Please see their Appendix B.1 for the descriptions of the environments.

**Experimental details**     Contrary to other settings, which report training episodes' return, we report performance in evaluation phases following Li et al. (2022). During evaluation phases, DA-AC uses discrete actions with the highest probability for the discrete components and mean actions for the continuous components. We use the implementations provided by Li et al. for baselines, including PADDPG (Hausknecht & Stone, 2016), PDQN (Xiong et al., 2018), and HHQN (Fu et al., 2019). All the baselines incorporate clipped double Q-learning from TD3, with PADDPG renamed to PATD3. The hyperparameters of DA-AC are adjusted according to those of the baselines (see Table 13). Since PDQN uses per-environment tuned $\gamma$ in Li et al., our results are slightly different than theirs as we use a fixed $\gamma = 0.99$ for PDQN to be consistent with other algorithms. Learning curves can be found in Figure 16.

### D.8   PAIRED $t$-TESTS FOR THE EFFECTIVENESS OF ICL

To assess the effectiveness of ICL, we conduct two-sided paired $t$-tests comparing DA-AC with and without ICL across benchmark settings. For each setting, tests are performed over paired seeds pooled across environments, where pairs share the same random seed. We additionally conduct a paired $t$-test on the mean performance differences aggregated across the seven settings. Table 1 reports the mean differences, $t$-statistics, $p$-values, and sample sizes. Overall, the results show that incorporating ICL yields statistically significant improvements in several settings and a positive mean improvement across settings.

Table 1: Paired $t$-tests comparing DA-AC and DA-AC w/o ICL across benchmark settings. For each setting, the test is conducted over paired seeds pooled across environments. The final row reports a paired $t$-test on the mean performance differences aggregated across the seven settings.

| Setting | Mean diff | $t$ | $p$ | $n$ |
|---|---|---|---|---|
| MuJoCo | 0.0760 | 3.306 | **0.0018** | 50 |
| DMC | 0.0157 | 1.115 | 0.2666 | 150 |
| Discretized MuJoCo | 0.1157 | 5.526 | $\mathbf{1.25 \times 10^{-6}}$ | 50 |
| Discretized DMC | 0.0383 | 2.149 | **0.0333** | 150 |
| Gym | 0.0510 | 1.570 | 0.1274 | 30 |
| MinAtar | 0.0158 | 0.409 | 0.6844 | 50 |
| PAMDPs | 0.0501 | 2.844 | **0.0059** | 70 |
| Across Settings | 0.0508 | 3.820 | **0.0088** | 7 |

### D.9 COMPUTATIONAL RESOURCE REQUIREMENT

All training for bandits was conducted on a local machine with AMD Ryzen 9 5900X 12-Core Processor. Each training run was executed using a single CPU core and consumed less than 256MB of RAM. Most runs completed 2000 training steps within 10 seconds.

All training for other tasks was conducted on CPU servers. These servers were equipped with a diverse range of Intel Xeon processors, including Intel E5-2683 v4 Broadwell @ 2.1GHz, Intel Platinum 8160F Skylake @ 2.1GHz, and Intel Platinum 8260 Cascade Lake @ 2.4GHz. Each training run was executed using a single CPU core and consumed less than 2GB of RAM. The training duration varied considerably across environments, primarily influenced by the dimensionality of the observation space, the complexity of the physics simulation, and, in the case of discrete action spaces, the dimensionality of the action space. For MuJoCo simulation tasks, most algorithms typically completed 1 million training steps in approximately 7 hours per run. However, LR-AC required a longer training period of roughly 9 hours due to the additional computational overhead of learning an extra neural network for computing the baseline.

Table 2: Observation and action dimensions of OpenAI Gym MuJoCo environments.

| Environment | Observation dimension | Action dimension |
|---|---|---|
| Hopper-v3 | 11 | 3 |
| Walker2d-v3 | 17 | 6 |
| HalfCheetah-v3 | 17 | 6 |
| Ant-v3 | 27 | 8 |
| Humanoid-v3 | 376 | 17 |

Table 3: Observation and action dimensions of DeepMind Control Suite environments.

| Domain | Task(s) | Observation dimension | Action dimension |
|---|---|---|---|
| pendulum | swingup | 3 | 1 |
| acrobot | swingup | 6 | 1 |
| reacher | hard | 6 | 2 |
| finger | turn_hard | 12 | 2 |
| hopper | stand, hop | 15 | 4 |
| fish | swim | 24 | 5 |
| swimmer | swimmer6 | 25 | 5 |
| cheetah | run | 17 | 6 |
| walker | run | 24 | 6 |
| quadruped | walk, run | 58 | 12 |
| humanoid | stand, walk, run | 67 | 24 |

# E HYPERPARAMETERS

Table 4: Hyperparameters for both continuous (col 3) and discrete (col 2) bandits that are different from Table 5. DA-AC is applied to both settings, denoted as DA-AC (C) and DA-AC (D), respectively. RP-AC uses the same hyperparameters as DA-AC (C); LR-AC uses the same hyperparameters as DA-AC (D).

| Hyperparameter | DA-AC (D) | DA-AC (C) |
|---|---|---|
| Batch size | 8 | |
| Learning rate (actor / critic) | 0.01 | |
| Neurons per hidden layer | (16, 16) | |
| Discount factor ($\gamma$) | N/A | |
| Replay buffer size | 2000 | |
| Policy update delay ($N_d$) | 1 | |
| Uniform exploration steps | N/A | |
| Learnable $\sigma$ range ($[\sigma_{\min}, \sigma_{\max}]$) | N/A | $[e^{-3}, e]$ |

Table 5: Hyperparameters of actor-critic algorithms for both MuJoCo/DMC continuous (cols 3–5) and discrete (col 2) control environments. DA-AC is applied to both settings, denoted as DA-AC (C) and DA-AC (D), respectively. For simplicity, we assume $[a_{\min}, a_{\max}] = [-1, 1]$ for continuous control algorithms. RP-AC uses the same hyperparameters as DA-AC (C); LR-AC and ST-AC use the same hyperparameters as DA-AC (D).

| Hyperparameter | DA-AC (D) | DA-AC (C) | TD3 | SAC |
|---|---|---|---|---|
| Batch size | 256 | | | |
| Optimizer | Adam | | | |
| Learning rate (actor / critic) | 0.0003 | | | 0.0003 / 0.001 |
| Target network update rate ($\tau$) | 0.005 | | | |
| Gradient steps per env step | 1 | | | |
| Number of hidden layers | 2 | | | |
| Neurons per hidden layer | (256, 256) | | | |
| Activation function | ReLU | | | |
| Discount factor ($\gamma$) | 0.99 | | | |
| Replay buffer size | $1 \times 10^6$ | | | |
| Policy update delay ($N_d$) | 2 | | | |
| Uniform exploration steps | 25,000 | | | 5,000 |
| Learnable $\sigma$ range ($[\sigma_{\min}, \sigma_{\max}]$) | N/A | $[0.05, 0.2]$ | N/A | N/A |
| Target entropy | N/A | | | $|\mathcal{A}|$ |
| Target policy noise clip ($c$) | N/A | | 0.5 | N/A |
| Target policy noise ($\tilde{\sigma}_{\text{TD3}}$) | N/A | | 0.2 | N/A |
| Exploration policy noise ($\sigma_{\text{TD3}}$) | N/A | | 0.1 | N/A |

Table 6: Hyperparameters of actor-critic algorithms for Gym environments that are different from Table 5. EAC, LR-AC, and ST-AC use the same hyperparameters as DA-AC.

| Hyperparameter | DA-AC | DSAC |
|---|---|---|
| Batch size | 128 | |
| Learning rate (actor / critic) | 0.0003 | |
| Target network update rate ($\tau$) | 0.01 | |
| Gradient steps per env step | 1 (every 10 env steps) | |
| Neurons per hidden layer | (120, 84) | |
| Replay buffer size | $1 \times 10^4$ | |
| Policy update delay ($N_d$) | 1 | |
| Uniform exploration steps | $12,500$ | $2,500$ |
| Target entropy | N/A | $0.89|\mathcal{A}|$ |

Table 7: Hyperparameters of actor-critic algorithms for MinAtar environments that are different from Table 5. EAC, LR-AC, and ST-AC use the same hyperparameters as DA-AC.

| Hyperparameter | DA-AC | DSAC |
|---|---|---|
| Batch size | 32 | |
| Learning rate (actor / critic) | 0.0003 | |
| Gradient steps per env step | 1 (every 4 env steps) | |
| Number of Conv. layers | 1 | |
| Conv. channels | 16 | |
| Conv. filter size | 3 | |
| Conv. stride | 1 | |
| Number of MLP layers | 2 | |
| Neurons per MLP layer | (128, 8) | |
| Replay buffer size | $1 \times 10^5$ | |
| Policy update delay ($N_d$) | 1 | |
| Uniform exploration steps | $50,000$ | $4,000$ |
| Target entropy | N/A | $0.89|\mathcal{A}|$ |

Table 8: Hyperparameters of PPO in MuJoCo/DMC continuous- and discrete-control environments.

| Hyperparameter | PPO |
|---|---|
| Optimizer | Adam |
| Learning rate (actor / critic) | $3 \times 10^{-4}$ |
| Discount factor ($\gamma$) | 0.99 |
| GAE parameter ($\lambda$) | 0.95 |
| Rollout length (timesteps per update) | 2048 |
| Minibatch size | 32 |
| Number of epochs per update | 10 |
| Number of hidden layers | 2 |
| Neurons per hidden layer | (64, 64) |
| Activation function | Tanh |
| Clipping parameter ($\epsilon$) | 0.2 |
| Entropy coefficient | 0.0 |
| Value loss coefficient | 0.5 |
| Max grad norm | 0.5 |
| Reward normalization | Enabled |
| Observation normalization | Enabled |
| Learning rate schedule | Linear decay |

Table 9: Hyperparameters of PPO in Gym environments that are different from Table 8.

| Hyperparameter | PPO |
|---|---|
| Rollout length (timesteps per update) | 128 |
| Number of epochs per update | 4 |

Table 10: Hyperparameters of PPO in MinAtar environments that are different from Table 8. Since MinAtar already normalizes the observations and rewards, we disable the normalization wrappers.

| Hyperparameter | PPO |
|---|---|
| Learning rate (actor / critic) | $2.5 \times 10^{-4}$ |
| Rollout length (timesteps per update) | 128 |
| Number of epochs per update | 4 |
| Number of Conv. layers | 1 |
| Conv. channels | 16 |
| Conv. filter size | 3 |
| Conv. stride | 1 |
| Number of MLP layers | 1 |
| Neurons per MLP layer | (128,) |
| Activation function | ReLU |
| Entropy coefficient | 0.01 |
| Reward normalization | Disable |
| Observation normalization | Disable |

Table 11: Hyperparameters of DQN in Gym environments.

| Hyperparameter | DQN |
|---|---|
| Batch size | 128 |
| Optimizer | Adam |
| Learning rate | $2.5 \times 10^{-4}$ |
| Discount factor ($\gamma$) | 0.99 |
| Hard target network update | Every 500 env steps |
| Gradient steps per env step | 1 (every 10 env steps) |
| Number of hidden layers | 2 |
| Neurons per hidden layer | (120, 84) |
| Activation function | ReLU |
| Replay buffer size | $1 \times 10^4$ |
| Min replay size before learning | 10,000 |
| Linear $\varepsilon$-greedy range | $1.0 \rightarrow 0.05$ |
| Linear $\varepsilon$-greedy steps | $2.5 \times 10^5$ |

Table 12: Hyperparameters of DQN in MinAtar environments.

| Hyperparameter | DQN |
|---|---|
| Batch size | 32 |
| Learning rate | $1 \times 10^{-4}$ |
| Hard target network update | Every 1000 env steps |
| Gradient steps per env step | 1 (every 4 env steps) |
| Number of Conv. layers | 1 |
| Conv. channels | 16 |
| Conv. filter size | 3 |
| Conv. stride | 1 |
| Number of MLP layers | 1 |
| Neurons per MLP layer | (128,) |
| Replay buffer size | $1 \times 10^5$ |
| Min replay size before learning | 40,000 |
| Linear $\varepsilon$-greedy range | $1.0 \rightarrow 0.01$ |
| Linear $\varepsilon$-greedy steps | $5 \times 10^5$ |

Table 13: Hyperparameters of actor-critic algorithms for PAMDP environments.

| Hyperparameter | DA-AC | PATD3 | HHQN | PDQN |
|---|---|---|---|---|
| Batch size | | 128 | | |
| Optimizer | | Adam | | |
| Learning rate (actor / critic) | | 0.0003 | | 0.0001 / 0.001 |
| Target network update rate ($\tau$) | | 0.005 | | 0.001 / 0.01 |
| Gradient steps per env step | | 1 | | |
| Number of hidden layers | | 2 | | |
| Neurons per hidden layer | | (256, 256) | | |
| Activation function | | ReLU | | |
| Discount factor ($\gamma$) | | 0.99 | | |
| Replay buffer size | | $1 \times 10^5$ | | |
| Policy update delay ($N_d$) | | 2 | | |
| Uniform exploration steps | 5,000 | N/A | | |
| Learnable $\sigma$ range ($[\sigma_{\min}, \sigma_{\max}]$) | [0.05, 0.2] | N/A | | |
| Target policy noise clip ($c$) | N/A | 0.5 | | N/A |
| Target policy noise ($\tilde{\sigma}_{\text{TD3}}$) | N/A | 0.2 | | N/A |
| Exploration policy noise ($\sigma_{\text{TD3}}$) | N/A | 0.1 | | N/A |
| Ornstein-Uhlenbeck noise | | N/A | | Enable |
| Linear $\varepsilon$-greedy range | | N/A | | $1.0 \rightarrow 0.01$ |
| Linear $\varepsilon$-greedy steps | | N/A | | $1 \times 10^3$ |
| Max grad norm | | N/A | | 0.5 |

# F ADDITIONAL PLOTS

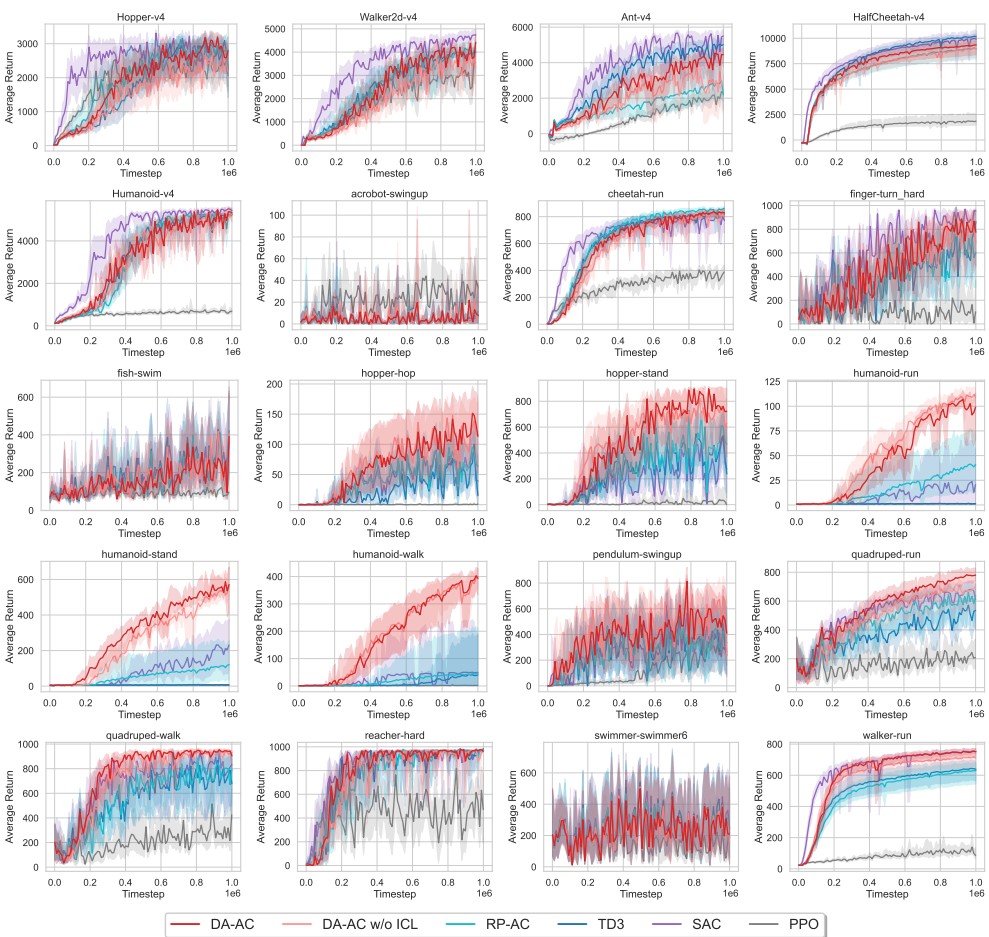

Figure 13: **Learning curves of DA-AC, DA-AC w/o ICL, and baselines** in 5 MuJoCo and 15 DMC *continuous* control tasks. Results are averaged over 10 seeds. Shaded regions show 95% bootstrap CIs.

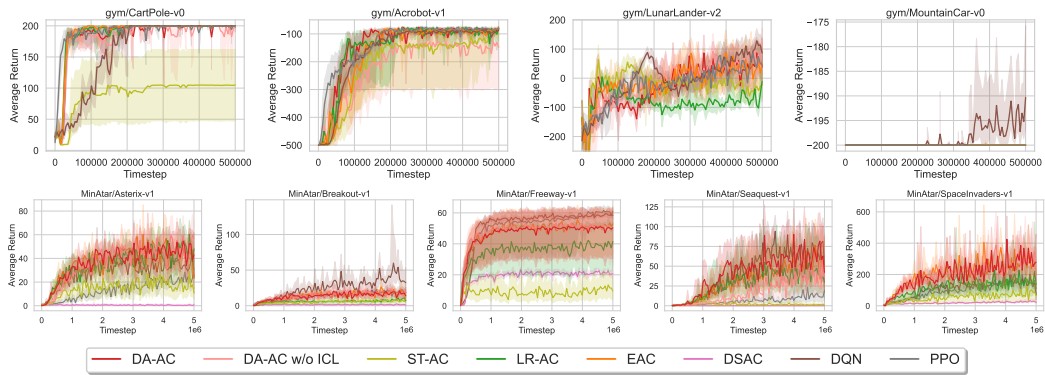

Figure 14: **Learning curves of DA-AC, DA-AC w/o ICL, and baselines** in 4 Gym and 5 MinAtar *discrete* control tasks. Results are averaged over 10 seeds. Shaded regions show 95% bootstrap CIs.

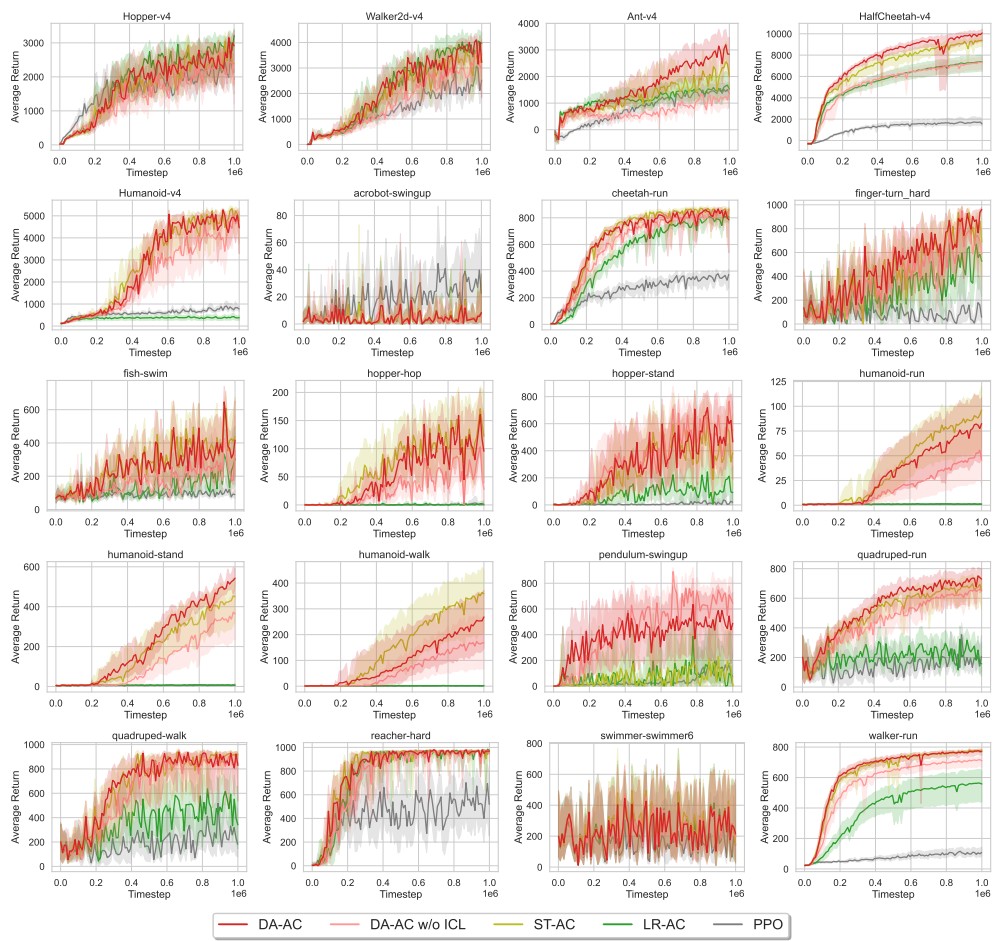

Figure 15: **Learning curves of DA-AC, DA-AC w/o ICL, and baselines** in 5 MuJoCo and 15 DMC *discrete* control tasks. Results are averaged over 10 seeds. Shaded regions show 95% bootstrap CIs.

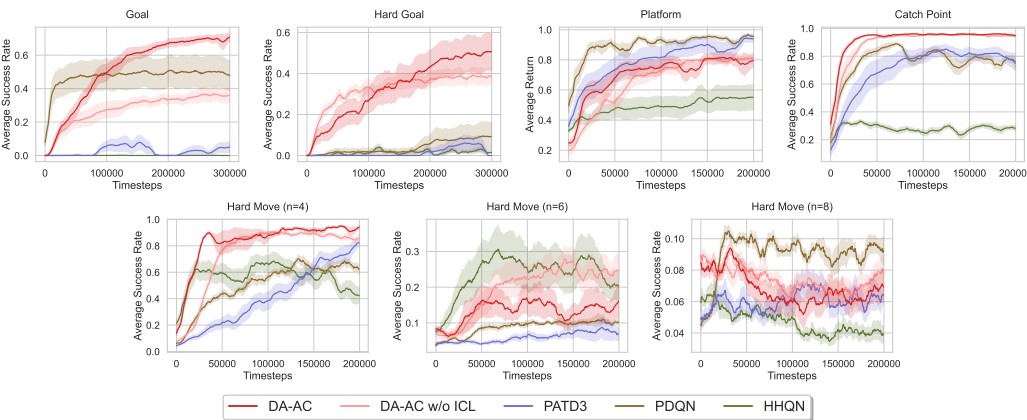

Figure 16: **Learning curves of DA-AC, DA-AC w/o ICL, and baselines** in 7 *hybrid* control tasks. Results are averaged over 10 seeds. Shaded regions show 95% bootstrap CIs.

# G PSEUDOCODE

## G.1 DA-AC: DISTRIBUTIONS-AS-ACTIONS ACTOR-CRITIC

---

**Algorithm 2** DA-AC for diverse action spaces

---

Input action sampling function $f : \mathcal{U} \to \Delta(\mathcal{A})$ (see Appendix D.1 for $f$ in different settings)
Initialize parameters $\mathbf{w}_1, \mathbf{w}_2, \boldsymbol{\theta}, \bar{\mathbf{w}}_1 \leftarrow \mathbf{w}_1, \bar{\mathbf{w}}_2 \leftarrow \mathbf{w}_2$, replay buffer $\mathcal{B}$
Obtain initial state $S_0$
**for** $t = 0$ to $T - 1$ **do**
    Take action $A_t \sim f(\cdot | U_t)$ with $U_t = \bar{\pi}_{\boldsymbol{\theta}}(S_t)$, observe $R_{t+1}, S_{t+1}$
    Add $\langle S_t, U_t, A_t, S_{t+1}, R_{t+1} \rangle$ to the buffer $\mathcal{B}$
    Sample a mini-batch $B$ from buffer $\mathcal{B}$
    Sample $\hat{U} = \omega U + (1 - \omega) U_A, \omega \sim \text{Uniform}[0, 1]$, for each transition $\langle S, U, A, S', R \rangle$ in $B$
    Update critics on $B$:

$$\mathbf{w}_i \leftarrow \mathbf{w}_i + \alpha_t \left( R + \gamma \min_{j \in \{1,2\}} Q_{\bar{\mathbf{w}}_j}(S', \bar{\pi}_{\boldsymbol{\theta}}(S')) - Q_{\mathbf{w}_i}(S, \hat{U}) \right) \nabla_{\mathbf{w}_i} Q_{\mathbf{w}_i}(S, \hat{U})$$

    **if** $t \equiv 0 \pmod{N_d}$ **then**
        Update policy on $B$:

$$\boldsymbol{\theta} \leftarrow \boldsymbol{\theta} - \alpha_t \nabla_{\boldsymbol{\theta}} \bar{\pi}_{\boldsymbol{\theta}}(S)^\top \nabla_{\tilde{U}} Q_{\mathbf{w}_1}(S, \tilde{U})|_{\tilde{U} = \bar{\pi}_{\boldsymbol{\theta}}(S)}$$

        Update target network weights:

$$\bar{\mathbf{w}}_i \leftarrow \tau \mathbf{w}_i + (1 - \tau) \bar{\mathbf{w}}_i$$

    **end if**
**end for**

---

## G.2 TD3: TWIN DELAYED DEEP DETERMINISTIC POLICY GRADIENT

---

**Algorithm 3** TD3 for continuous action spaces

---

Input exploration noise $\sigma_{\text{TD3}}$, target policy noise $\tilde{\sigma}_{\text{TD3}}$, target noise clipping $c$
Initialize parameters $\mathbf{w}_1, \mathbf{w}_2, \boldsymbol{\theta}, \bar{\mathbf{w}}_1 \leftarrow \mathbf{w}_1, \bar{\mathbf{w}}_2 \leftarrow \mathbf{w}_2, \bar{\boldsymbol{\theta}} \leftarrow \boldsymbol{\theta}$, replay buffer $\mathcal{B}$
Obtain initial state $S_0$
**for** $t = 0$ to $T - 1$ **do**
    Take action $A_t = \pi_{\boldsymbol{\theta}}(S_t) + \epsilon, \epsilon \sim \mathcal{N}(0, \sigma_{\text{TD3}})$, and observe $R_{t+1}, S_{t+1}$
    Add $\langle S_t, A_t, S_{t+1}, R_{t+1} \rangle$ to the buffer $\mathcal{B}$
    Sample a mini-batch $B$ from buffer $\mathcal{B}$
    Sample $A' = \pi_{\bar{\boldsymbol{\theta}}}(S') + \epsilon, \epsilon \sim \text{clip}(\mathcal{N}(0, \tilde{\sigma}_{\text{TD3}}), -c, c)$, for each transition $\langle S, A, S', R \rangle$ in $B$
    Update critics on $B$:

$$\mathbf{w}_i \leftarrow \mathbf{w}_i + \alpha_t \left( R + \gamma \min_{j \in \{1,2\}} Q_{\bar{\mathbf{w}}_j}(S', A') - Q_{\mathbf{w}_i}(S, A) \right) \nabla_{\mathbf{w}_i} Q_{\mathbf{w}_i}(S, A)$$

    **if** $t \equiv 0 \pmod{N_d}$ **then**
        Update policy on $B$:

$$\boldsymbol{\theta} \leftarrow \boldsymbol{\theta} - \alpha_t \nabla_{\boldsymbol{\theta}} \pi_{\boldsymbol{\theta}}(S)^\top \nabla_{\tilde{A}} Q_{\mathbf{w}_1}(S, \tilde{A})|_{\tilde{A} = \pi_{\boldsymbol{\theta}}(S)}$$

        Update target network weights:

$$\bar{\mathbf{w}}_i \leftarrow \tau \mathbf{w}_i + (1 - \tau) \bar{\mathbf{w}}_i, \quad \bar{\boldsymbol{\theta}} \leftarrow \tau \boldsymbol{\theta} + (1 - \tau) \bar{\boldsymbol{\theta}}$$

    **end if**
**end for**

---

## G.3 RP-AC: ACTOR-CRITIC WITH THE REPARAMETERIZATION (RP) ESTIMATOR

---

**Algorithm 4** RP-AC for continuous action spaces

---

Input reparameterization function $g_{\boldsymbol{\theta}} : \mathcal{S} \times \mathbb{R} \to \mathcal{A}$ (for Gaussian policies, see Appendix D.1)
Initialize parameters $\mathbf{w}_1, \mathbf{w}_2, \boldsymbol{\theta}, \bar{\mathbf{w}}_1 \leftarrow \mathbf{w}_1, \bar{\mathbf{w}}_2 \leftarrow \mathbf{w}_2$, replay buffer $\mathcal{B}$
Obtain initial state $S_0$
**for** $t = 0$ to $T - 1$ **do**
    Take action $A_t = g_{\boldsymbol{\theta}}(\epsilon; S_t)$, $\epsilon \sim \mathcal{N}(0, 1)$, and observe $R_{t+1}, S_{t+1}$
    Add $\langle S_t, A_t, S_{t+1}, R_{t+1} \rangle$ to the buffer $\mathcal{B}$
    Sample a mini-batch $B$ from buffer $\mathcal{B}$
    Sample $A' = g_{\boldsymbol{\theta}}(\epsilon; S')$, $\epsilon \sim \mathcal{N}(0, 1)$, for each transition $\langle S, A, S', R \rangle$ in $B$
    Update critics on $B$:

$$\mathbf{w}_i \leftarrow \mathbf{w}_i + \alpha_t \left( R + \gamma \min_{j \in \{1,2\}} Q_{\bar{\mathbf{w}}_j}(S', A') - Q_{\mathbf{w}_i}(S, A) \right) \nabla_{\mathbf{w}_i} Q_{\mathbf{w}_i}(S, A)$$

    **if** $t \equiv 0 \pmod{N_d}$ **then**
        Sample $\epsilon \sim \mathcal{N}(0, 1)$ for each transition $\langle S, A, S', R \rangle$ in $B$
        Update policy on $B$:

$$\boldsymbol{\theta} \leftarrow \boldsymbol{\theta} - \alpha_t \nabla_{\boldsymbol{\theta}} g_{\boldsymbol{\theta}}(\epsilon; S)^{\top} \nabla_{\tilde{A}} Q_{\mathbf{w}_1}(S, \tilde{A})|_{\tilde{A} = g_{\boldsymbol{\theta}}(\epsilon; S)}$$

        Update target network weights:

$$\bar{\mathbf{w}}_i \leftarrow \tau \mathbf{w}_i + (1 - \tau) \bar{\mathbf{w}}_i$$

    **end if**
**end for**

---

## G.4 ST-AC: ACTOR-CRITIC WITH THE STRAIGHT-THROUGH (ST) ESTIMATOR

---

**Algorithm 5** ST-AC for discrete action spaces

---

Initialize parameters $\mathbf{w}_1, \mathbf{w}_2, \boldsymbol{\theta}, \bar{\mathbf{w}}_1 \leftarrow \mathbf{w}_1, \bar{\mathbf{w}}_2 \leftarrow \mathbf{w}_2$, replay buffer $\mathcal{B}$
Obtain initial state $S_0$
**for** $t = 0$ to $T - 1$ **do**
    Take action $A_t \sim \pi_{\boldsymbol{\theta}}(\cdot|S_t)$, and observe $R_{t+1}, S_{t+1}$
    Add $\langle S_t, A_t, S_{t+1}, R_{t+1} \rangle$ to the buffer $\mathcal{B}$
    Sample a mini-batch $B$ from buffer $\mathcal{B}$
    Sample $A' \sim \pi_{\boldsymbol{\theta}}(\cdot|S')$ for each transition $\langle S, A, S', R \rangle$ in $B$
    Update critics on $B$:

$$\mathbf{w}_i \leftarrow \mathbf{w}_i + \alpha_t \left( R + \gamma \min_{j \in \{1,2\}} Q_{\bar{\mathbf{w}}_j}(S', A') - Q_{\mathbf{w}_i}(S, A) \right) \nabla_{\mathbf{w}_i} Q_{\mathbf{w}_i}(S, A)$$

    **if** $t \equiv 0 \pmod{N_d}$ **then**
        Sample $\tilde{A} \sim \pi_{\boldsymbol{\theta}}(\cdot|S)$, for each transition $\langle S, A, S', R \rangle$ in $B$
        Use the straight-through trick to compute $\tilde{A}_{\boldsymbol{\theta}} = \text{one\_hot}(\tilde{A}) + \pi_{\boldsymbol{\theta}}(\cdot|S) - \pi_{\boldsymbol{\phi}}(\cdot|S)|_{\boldsymbol{\phi} = \boldsymbol{\theta}}$
        Update policy on $B$:

$$\boldsymbol{\theta} \leftarrow \boldsymbol{\theta} - \alpha_t \nabla_{\boldsymbol{\theta}} \pi_{\boldsymbol{\theta}}(\cdot|S)^{\top} \nabla_{\tilde{A}} Q_{\mathbf{w}_1}(S, \tilde{A})|_{\tilde{A} = \tilde{A}_{\boldsymbol{\theta}}}$$

        Update target network weights:

$$\bar{\mathbf{w}}_i \leftarrow \tau \mathbf{w}_i + (1 - \tau) \bar{\mathbf{w}}_i$$

    **end if**
**end for**

---

## G.5 LR-AC: Actor-critic with the likelihood-ratio (LR) estimator

---

**Algorithm 6** LR-AC for diverse action spaces

---

Initialize parameters $\mathbf{w}_1, \mathbf{w}_2, \boldsymbol{\theta}, \mathbf{v}, \bar{\mathbf{w}}_1 \leftarrow \mathbf{w}_1, \bar{\mathbf{w}}_2 \leftarrow \mathbf{w}_2$, replay buffer $\mathcal{B}$

Obtain initial state $S_0$

**for** $t = 0$ to $T - 1$ **do**

    Take action $A_t \sim \pi_{\boldsymbol{\theta}}(\cdot|S_t)$, and observe $R_{t+1}$, $S_{t+1}$

    Add $\langle S_t, A_t, S_{t+1}, R_{t+1} \rangle$ to the buffer $\mathcal{B}$

    Sample a mini-batch $B$ from buffer $\mathcal{B}$

    Sample $\tilde{A} \sim \pi_{\boldsymbol{\theta}}(\cdot|S)$ and $A' \sim \pi_{\boldsymbol{\theta}}(\cdot|S')$ for each transition $\langle S, A, S', R \rangle$ in $B$

    Update critics on $B$:

$$\mathbf{w}_i \leftarrow \mathbf{w}_i + \alpha_t \left( R + \gamma \min_{j \in \{1,2\}} Q_{\bar{\mathbf{w}}_j}(S', A') - Q_{\mathbf{w}_i}(S, A) \right) \nabla_{\mathbf{w}_i} Q_{\mathbf{w}_i}(S, A)$$

$$\mathbf{v} \leftarrow \mathbf{v} + \alpha_t \left( Q_{\mathbf{w}_1}(S, \tilde{A}) - V_{\mathbf{v}}(S) \right) \nabla_{\mathbf{v}} V_{\mathbf{v}}(S)$$

    **if** $t \equiv 0 \pmod{N_d}$ **then**

        Sample $\tilde{A} \sim \pi_{\boldsymbol{\theta}}(\cdot|S)$, for each transition $\langle S, A, S', R \rangle$ in $B$

        Update policy on $B$:

$$\boldsymbol{\theta} \leftarrow \boldsymbol{\theta} - \alpha_t \nabla_{\boldsymbol{\theta}} \log \pi_{\boldsymbol{\theta}}(\tilde{A}|S) \left( Q_{\mathbf{w}_1}(S, \tilde{A}) - V_{\mathbf{v}}(S) \right)$$

        Update target network weights:

$$\bar{\mathbf{w}}_i \leftarrow \tau \mathbf{w}_i + (1 - \tau) \bar{\mathbf{w}}_i$$

    **end if**

**end for**

---

## G.6 EAC: Actor-critic with the expected policy gradient estimator

---

**Algorithm 7** EAC for discrete action spaces

---

Initialize parameters $\mathbf{w}_1, \mathbf{w}_2, \bar{\mathbf{w}}_1 \leftarrow \mathbf{w}_1, \bar{\mathbf{w}}_2 \leftarrow \mathbf{w}_2$, replay buffer $\mathcal{B}$

Obtain initial state $S_0$

**for** $t = 0$ to $T - 1$ **do**

    Take action $A_t \sim \pi_{\boldsymbol{\theta}}(\cdot|S_t)$, and observe $R_{t+1}$, $S_{t+1}$

    Add $\langle S_t, A_t, S_{t+1}, R_{t+1} \rangle$ to the buffer $\mathcal{B}$

    Sample a mini-batch $B$ from buffer $\mathcal{B}$

    Sample $\tilde{A} \sim \pi_{\boldsymbol{\theta}}(\cdot|S)$ and $A' \sim \pi_{\boldsymbol{\theta}}(\cdot|S')$ for each transition $\langle S, A, S', R \rangle$ in $B$

    Update critics on $B$:

$$\mathbf{w}_i \leftarrow \mathbf{w}_i + \alpha_t \left( R + \gamma \min_{j \in \{1,2\}} Q_{\bar{\mathbf{w}}_j}(S', A') - Q_{\mathbf{w}_i}(S, A) \right) \nabla_{\mathbf{w}_i} Q_{\mathbf{w}_i}(S, A)$$

    **if** $t \equiv 0 \pmod{N_d}$ **then**

        Update policy on $B$:

$$\boldsymbol{\theta} \leftarrow \boldsymbol{\theta} - \alpha_t \nabla_{\boldsymbol{\theta}} \sum_{a \in \mathcal{A}} \pi_{\boldsymbol{\theta}}(a|S) Q_{\mathbf{w}_1}(S, a)$$

        Update target network weights:

$$\bar{\mathbf{w}}_i \leftarrow \tau \mathbf{w}_i + (1 - \tau) \bar{\mathbf{w}}_i$$

    **end if**

**end for**

---

## H    USE OF LARGE LANGUAGE MODELS

Large language models (LLMs) were employed in a strictly auxiliary capacity during the preparation of this paper. Their use was limited to two areas: (1) assisting with writing refinement by improving readability, grammar, and conciseness, without contributing to the technical content or conceptual development; and (2) supporting workflow tasks such as drafting or adjusting scripts for data processing and figure generation, with all outputs carefully reviewed and corrected by the authors. LLMs were not used for generating research ideas, conducting literature searches, or producing original technical material. Their involvement was confined to polishing communication and light implementation support.

