# OpenReview forum: "Distributions as Actions: A Unified Framework for Diverse Action Spaces"
_ICLR.cc/2026/Conference — ICLR 2026 Poster_

### Official Review · Reviewer_Xv1R · 2025-10-17

**Soundness:** 3
**Presentation:** 3
**Contribution:** 3
**Rating:** 6
**Confidence:** 4

**Summary:**

This paper introduces a framework for reinforcement learning called "distributions-as-actions". The authors try to redefine the boundary between the agent and the environment. Instead of the agent outputting an action from a potentially discrete or hybrid space, the agent outputs the continuous parameters of a probability distribution over actions (e.g., the probability vector for a categorical policy, or the mean and variance for a Gaussian policy). The environment is then responsible for sampling the final action from this distribution. This reparameterization transforms any RL problem into one with a continuous action space, allowing for the development of unified algorithms.

The authors derive a corresponding policy gradient estimator. To address the challenge of learning a critic over this new, more complex action space, they propose a simple and effective technique called Interpolated Critic Learning (ICL). Finally, they combine these ideas into a practical actor-critic algorithm, DA-AC, based on td3.

**Strengths:**

1. The core idea of treating distribution parameters as actions is a simple and interesting concept. It unifies diverse action spaces (discrete, continuous, hybrid) into a single, continuous control problem, potentially simplifying algorithm design
2. The introduction of Interpolated Critic Learning (ICL) is a practical solution to the challenge of learning the critic in this new, higher-dimensional parameter space
3. The authors test their method on over 30 tasks across continuous, discrete, high-dimensional discrete, and hybrid domains, consistently showing competitive performance

**Weaknesses:**

1. The paper presents the transformation from discrete to continuous action spaces as a clear benefit, but does not sufficiently discuss the potential downsides. Discrete action spaces have a specific structure that is often leveraged by advanced algorithms. For instance:
- Entropy regularization is commonly used in RL for promoting exploration in discrete spaces. It is unclear how an equivalent and effective exploration strategy would be implemented in the continuous parameter space. Simply encouraging entropy over the parameters (a vector of probabilities) is not the same as encouraging entropy in the resulting categorical distribution.
- Methods like mirror descent are well-suited for optimization over the probability simplex. By moving to an unconstrained Euclidean space of parameters, these geometric advantages are lost.
- Planning algorithms like MCTS are fundamentally built on discrete action branching. The proposed framework makes it difficult to see how such methods could be integrated. A deeper discussion of what is "lost" in this transformation and how the framework might address these challenges would significantly strengthen the paper.
2. The paper should better position itself with respect to other works that also re-frame RL problems through representation learning. For example, "Representation-Driven RL" (Nabati et al.)  also learns a latent space for policies and uses bandit algorithms to guide exploration in that space. While the mechanism is different (RepRL learns a representation of the entire policy, whereas this paper redefines the action at each step), the high-level goal of shifting the problem into a more structured latent space is shared. Discussing these connections would provide a richer context for the paper's contribution.
3. The comparison to PPO, a very strong on-policy baseline should be more convincing. PPO's performance is known to be highly dependent on implementation details and computational budget (e.g., total environment steps, number of parallel environments). Without a more detailed analysis or comparison against a highly-tuned PPO baseline known to achieve SOTA on these tasks, it is difficult to be certain that DA-AC's superior performance isn't an artifact of a weakly-scaled baseline. A stronger claim could be made by showing that DA-AC is more sample-efficient or reaches a higher asymptotic performance in a "computationally unconstrained" setting where both algorithms are given unconstrained resources.

**Questions:**

- Could the authors elaborate on how entropy regularization could be effectively adapted to the distributions-as-actions framework?
- How does one encourage diverse actions by acting in the continuous parameter space of that distribution?
- What are the authors' thoughts on the compatibility of this framework with planning algorithms like MCTS, which rely on a discrete branching factor?
- Does this reframing inherently limit the applicability to model-free actor-critic methods?
- Regarding the PPO comparisons, can the authors provide more details on the implementation and tuning? For example, were the reported results for PPO based on a single-environment setup, or was it scaled with parallel environments as is common for achieving top performance?
- The ICL technique uses linear interpolation between the policy's output parameters $U_t$ and the deterministic parameters $U_{A_t}$. Could you provide more intuition on this choice? For categorical policies where the parameter space is a simplex, does this linear interpolation respect the geometry of the space, or does it operate on the unconstrained logits

---

> ### Author Response · Authors · 2025-11-21
> **Rebuttal I**
>
> We thank the reviewer for their careful reading of our paper and for the constructive feedback. We address all raised concerns in detail below.
>
> > **Q1**: Could the authors elaborate on how entropy regularization could be effectively adapted to the distributions-as-actions framework?
>
> **Response**: Thank you for the thoughtful question. Entropy regularization can indeed be incorporated into our distributions-as-actions (DA) framework, and there are two natural ways to do so.
>
> 1. **Entropy as part of the regularized objective.** In entropy-regularized actor–critic methods (e.g., SAC), the entropy term is added to the actor objective and can also be included in the critic target. In our setting, the policy outputs distribution parameters $U_t$, and the entropy of the induced action distribution $\mathcal{H}(f(\cdot | U_t))$ (or a sample-based term, $-\log f(A_t | U_t)$) can be computed and incorporated into both the actor and, if desired, the critic objective in a manner analogous to SAC.
> The critic can be trained using either the standard update (Eq. 12) or the interpolated critic (Eq. 14), with or without an entropy term in the TD target. The actor is then optimized using the distribution-parameter policy gradient (Eq. 11) augmented with the entropy regularization term. This preserves the role of entropy in encouraging exploration while naturally extending to the DA setting.
>
> 2. **Entropy absorbed into the reward transformation.** Alternatively, entropy can be treated as a reward-shaping term and incorporated directly into the transformed MDP. This corresponds to modifying the reward as $ \tilde R_{t+1}=R_{t+1}+\alpha \mathcal{H} ( f( \cdot | U_t)) $ or $\tilde R_{t+1} = R_{t+1} - \alpha \log f(A_t | U_t)$, after which the DA-PG update (Eq. 11) optimizes the shaped objective without altering the actor update rule. This leverages the modularity of our framework and keeps the optimization pipeline unchanged.
>
> Both approaches are compatible with DA-AC, and exploring these integration strategies in depth is an interesting direction for future work. We have clarified this connection in the added Appendix B in the updated version.
>
> > **Q2**: How does one encourage diverse actions by acting in the continuous parameter space of that distribution?
>
> **Response**: Thank you for the insightful question. Encouraging diverse actions in the DA framework is achieved by shaping the distribution parameters $u$, which determine the stochasticity of the induced action distribution $f(\cdot | u)$. There are several ways this can be done:
>
> 1. **Entropy regularization.** As discussed in our response to Q1, entropy regularization can be incorporated into the DA objective. Adding $\mathcal{H}(f(\cdot | U_t))$ (or $-\log f(A_t | U_t)$) to the actor loss directly encourages the policy to output parameters that yield broader or more exploratory action distributions. This extends existing entropy-based exploration methods naturally into the DA setting.
>
> 2. **Constraining the distribution-parameter space.** The DA framework allows structural constraints on the parameter space $\mathcal{U}$ to enforce minimum stochasticity:
>    - **Continuous actions**: bounding the standard deviation of a Gaussian (as done in DA-AC (Continuous)) ensures non-zero variance.
>    - **Discrete actions**: constraining the logits (e.g., via clipping or temperature limits) prevents premature collapse onto a single action.
>    These constraints encourage persistent exploration regardless of the specific value function landscape.
>
> 3. **Architectural or parameterization choices.** Different parameterizations of $f(\cdot | u)$ (e.g., mixture distributions or temperature-controlled softmax functions) can influence the diversity of sampled actions. Such choices could shape the sensitivity of the policy to multimodal or sparse-reward environments and potentially promote richer exploration.
>
> Overall, encouraging diverse actions in the DA framework is straightforward. While we have only explored Option 2 in the continuous action case, it’d be interesting to explore other options including those listed above.

---

> ### Author Response · Authors · 2025-11-21
> **Rebuttal II**
>
> > **Q3 & Q4**:
> (Q3) What are the authors' thoughts on the compatibility of this framework with planning algorithms like MCTS, which rely on a discrete branching factor?
> (Q4) Does this reframing inherently limit the applicability to model-free actor-critic methods?
>
> **Response**: Thank you for the interesting questions. To answer Q4 first, the distributions-as-actions (DA) reframing does only inherently limit the applicability to model-free methods. To answer Q3, while DA is not directly compatible with *vanilla discrete MCTS* as it produces continuous instead of discrete parameters $u$, **DA can be combined with planning algorithms including MCTS** in several practical ways:
>
> 1. The distribution-parameter space $\mathcal{U}$ can be discretized into a small set of representative parameter vectors, allowing vanilla discrete MCTS to operate over a finite set of “distributional actions.”
> 2. MCTS itself has been extended to allow for continuous actions (see Yee et al. (2016)), using progressive widening or sampling-based expansions. These methods could be directly applied under the DA framework.
> 3. In discrete-action environments, one may still run MCTS over the primitive discrete actions while using DA for value estimation.
> 4. Further, there exist other planning algorithms that are demonstrated to be effective with continuous action spaces like TD-MPC (Hansen et al., 2022) or model-based reparameterization policy gradient (Zhang et al., 2023), which could potentially be used in the DA framework.
>
> We view integrating DA with planning methods such as continuous-action MCTS as a promising direction for future work. We have included a discussion on the potential of model-based planning algorithms in Appendix B.
>
> Yee, et al. (2016). Monte Carlo Tree Search in Continuous Action Spaces with Execution Uncertainty. IJCAI.
>
> Hansen, et al. (2022). Temporal Difference Learning for Model Predictive Control. ICML.
>
> Zhang, et al. (2023). Model-based reparameterization policy gradient methods: Theory and practical algorithms. NeurIPS.
>
> > **Q5 & W3**:
> (Q5) ... can the authors provide more details on the implementation and tuning (of PPO)? For example, were the reported results for PPO based on a single-environment setup, or was it scaled with parallel environments as is common for achieving top performance?
> (W3) The comparison to PPO, a very strong on-policy baseline should be more convincing. ... Without a more detailed analysis or comparison against a highly-tuned PPO baseline known to achieve SOTA on these tasks, it is difficult to be certain that DA-AC's superior performance isn't an artifact of a weakly-scaled baseline. A stronger claim could be made by showing that DA-AC is more sample-efficient or reaches a higher asymptotic performance in a "computationally unconstrained" setting where both algorithms are given unconstrained resources.
>
> **Response**: Thank you for the helpful comments. We agree that PPO performance is highly sensitive to implementation details, parallelization, and compute budget. In this work, we focus on developing the first algorithm under the proposed framework starting with the simple but fundamental *single-stream setting*, so we use the original single stream PPO (Schulman et al., 2017) as a baseline. Extending DA-AC to the multi-stream setting (similar to how TD3 is extended in Seo et al., 2025) and comparing it with multi-stream PPO will be an interesting future work.
>
> Regarding the implementation details, we use the standard CleanRL implementation (Huang et al., 2022). For continuous actions, the hyperparameters are tuned in MuJoCo environments (Huang et al., 2024). Its performance is consistent with the original PPO paper (Schulman et al., 2017) and later works that compared with it including SAC (Haarnoja et al., 2018) and TD3 (Fujimoto et al., 2018). For discrete actions, the hyperparameters are also tuned by Huang et al. (2024) for gym discrete environments.
>
> We have clarified the single-stream setup in Section 4.4 and the version of PPO in Section 5.1 in the updated version. The specific details about the PPO’s implementation and hyperparameters are already presented in the original submission (see Appendices D.4, D.5 and Tables 7-9 in the updated version for details).
>
> Schulman, et al. (2017). Proximal policy optimization algorithms. arXiv preprint arXiv:1707.06347.
>
> Seo, et al. (2025). FastTD3: Simple, Fast, and Capable Reinforcement Learning for Humanoid Control. arXiv preprint arXiv:2505.22642.
>
> Haarnoja, et al. (2018). Soft actor-critic: Off-policy maximum entropy deep reinforcement learning with a stochastic actor. ICML.
>
> Fujimoto, et al. (2018). Addressing function approximation error in actor-critic methods. ICML.
>
> Huang, et al. (2022). CleanRL: High-quality single-file implementations of deep reinforcement learning algorithms. JMLR.
>
> Huang, et al. (2024). Open RLl benchmark: Comprehensive tracked experiments for reinforcement learning. arXiv preprint arXiv:2402.03046.

---

> ### Author Response · Authors · 2025-11-21
> **Rebuttal III**
>
> > **Q6**: The ICL technique uses linear interpolation between the policy's output parameters $U_{t}$ and the deterministic parameters $U_{A_t}$. Could you provide more intuition on this choice? For categorical policies where the parameter space is a simplex, does this linear interpolation respect the geometry of the space, or does it operate on the unconstrained logits
>
> **Response**: Thank you for the thoughtful question. The intuition behind ICL is to encourage the critic to learn smoother and more informative curvature along the path connecting the policy’s current distribution parameters $U_t$ and the deterministic parameters $U_{A_t}$ associated with the sampled action. The choice of $U_t$ is natural since $A_t$ is sampled from $f(\cdot | U_t)$. The choice of $U_{A_t}$ has two motivations: (1) learning the value at a deterministic distribution is often easier because its target is less noisy, and (2) the optimal parameter $U^*$ must correspond to $U_a$ for some $a \in \mathcal{A}$. By training the critic at interpolated points $\hat U_t = \omega U_t + (1-\omega) U_{A_t}$, we encourage the value surface to vary smoothly along behaviorally meaningful directions, resulting in more informative gradients for the actor (as illustrated in Figure 2).
>
> For categorical policies, the interpolation is performed directly in the **probability simplex**, i.e., between probability vectors (not logits), which naturally preserves the geometry of the space. Because the simplex is convex, the interpolated point $\hat U_t$ remains a valid categorical distribution, respecting the geometry of the space. This avoids any need for projection and ensures that both endpoints and intermediate points correspond to meaningful distributions. Empirically, we find that interpolating in the simplex yields stable training and improves the critic’s curvature in discrete-action settings (see Figures 2 and 8).
>
> In summary: (i) interpolation is chosen to shape the critic’s value landscape along directions relevant for policy improvement, and (ii) for categorical policies, interpolation is performed in the probability simplex, which preserves the geometry of the space.
>
> > **W1**: The paper presents the transformation from discrete to continuous action spaces as a clear benefit, but does not sufficiently discuss the potential downsides. Discrete action spaces have a specific structure that is often leveraged by advanced algorithms.
> > For instance: compatibility with entropy regularization, mirror descent, and MCTS.
>
> **Response**: Thank you for pointing this out. We have added a discussion on the trade-off in Appendix B, including a suggestion to use a hybrid approach if one wants to explicitly take advantage of structure like discrete actions. Here, we would like to clarify, though, that moving to a continuous action representation does not prevent the use of entropy regularization (see the response to Q1) and mirror descent (see Tomar et al. (2020) and Vaswani et al. (2022) for mirror descent’s application on continuous action spaces). As discussed in the response to Q3&Q4, we do lose the ability to apply MCTS in the standard form directly, but there are practical ways to use MCTS variants and other planning algorithms.
>
> Tomar, et al. (2020). Mirror Descent Policy Optimization. ICLR.
>
> Vaswani, et al. (2022). A general class of surrogate functions for stable and efficient reinforcement learning. ICML.
>
> > **W2**: The paper should better position itself with respect to other works that also re-frame RL problems through representation learning. For example, "Representation-Driven RL" (Nabati et al.) also learns a latent space for policies and uses bandit algorithms to guide exploration in that space. While the mechanism is different (RepRL learns a representation of the entire policy, whereas this paper redefines the action at each step), the high-level goal of shifting the problem into a more structured latent space is shared. Discussing these connections would provide a richer context for the paper's contribution.
>
> **Response**: We appreciate the reviewer’s comment. We have included an extended related work discussion in Appendix A, including representation-driven RL.
>
> We appreciate the reviewer’s time and constructive feedback. We hope our responses clarify the key points and demonstrate the potential of our contributions.

---

> > ### Comment · Reviewer_Xv1R · 2025-11-27
> >
> > I appreciate the reviewers thorough response. You've addressed most of my concerns. Please try to emphasize these tradeoffs and weaknesses in the paper. I believe you get another page for camera ready, it'd be nice to see a more thorough discussion of tradeoff and weaknesses of using this form of continuous action space.
> >
> > I think this paper merits publication in the conference and I'm raising my score. Good luck!

---

> > > ### Author Response · Authors · 2025-11-27
> > >
> > > Thank you for your response and for adjusting the rating. We truly appreciate your constructive feedback. We will certainly utilize the additional page to include a discussion of the tradeoffs and enhance the contextualization of our work. Thanks again!

---

### Official Review · Reviewer_WRJw · 2025-10-29

**Soundness:** 3
**Presentation:** 3
**Contribution:** 3
**Rating:** 6
**Confidence:** 3

**Summary:**

The paper proposes a unified framework for handling both discrete and continuous action spaces, which traditionally require distinct design choices in reinforcement learning. Instead of directly learning over the original action space defined by the MDP, the method learns policies and value functions over the distribution parameters from which actions are sampled. Because distribution parameters are typically continuous, both discrete and continuous action spaces can be handled in a unified manner. Central to the paper is the introduction of a gradient estimator defined on these parameters that exhibits lower variance but higher bias. This bias issue is effectively mitigated through an interpolated critic learning mechanism. Taken together, these advantages appear to be the primary factor driving the strong empirical performance of the proposed approach across a range of tasks.

The work evaluates the method on tasks with diverse action space types and demonstrates consistent performance advantages.

**Strengths:**

The paper is well-structured and comprehensive. While the idea may appear straightforward at first glance, the authors provide thorough motivation, analysis, implementation details, and experimental results, which I find convincing.

**Weaknesses:**

Based on my reading, the paper does not include a dedicated related work section where the connection between this approach and prior research is explicitly discussed. While the authors do reference commonly used policy gradients in Section 2 and baseline algorithms in the experiments, I would be interested in a deeper discussion of previous efforts that also attempt to learn policy distribution parameters—perhaps separately in continuous and discrete action spaces. Such a contextualization would help better highlight and clarify the contributions of the proposed method.

**Questions:**

As noted in the weaknesses section, I would encourage the authors to provide a more explicit discussion of how this work relates to prior literature.

---

> ### Author Response · Authors · 2025-11-21
> **Rebuttal**
>
> We thank the reviewer for their helpful comments. We address the concern on the contextualization of our work below.
>
> > **W1 & Q1**: (W1) Based on my reading, the paper does not include a dedicated related work section where the connection between this approach and prior research is explicitly discussed. While the authors do reference commonly used policy gradients in Section 2 and baseline algorithms in the experiments, I would be interested in a deeper discussion of previous efforts that also attempt to learn policy distribution parameters—perhaps separately in continuous and discrete action spaces. Such a contextualization would help better highlight and clarify the contributions of the proposed method. (Q1) I would encourage the authors to provide a more explicit discussion of how this work relates to prior literature.
>
> **Response**: An alternative to a dedicated Related Work section is to include related work in the introduction, to clearly place our proposed approach. However, we acknowledge that we can go more deeply into related work. We have included an extended related work discussion in Appendix A that covers how policy distribution parameters are updated in both continuous and discrete settings as well as other related works.
>
> We believe our response addresses the reviewers’ concern. We appreciate their time invested in evaluating our work.

---

### Official Review · Reviewer_8RUo · 2025-10-29

**Soundness:** 4
**Presentation:** 4
**Contribution:** 4
**Rating:** 8
**Confidence:** 5

**Summary:**

Distributions as actions is a new way for representing the policy.
Instead of predicting an action, the policy predicts the action distribution.

This work then presents how to integrate this into the env (the env has a sampler function) and into the critic (the critic estimates V of the distribution) and how to train the critic (since next-states arrive from sampling from the distribution and then simulating).

**Strengths:**

DA-AC seems like a very promising framework.
The work itself is interesting and the concept is novel.
Results show it outperforms pre-existing methods on simple tasks.

**Weaknesses:**

While the authors show the interpolated critic is an important feature and very nicely present how this helps the critic learn a better optimization landscape, it isn't clear what part of the agent is benefitting from this method.

When comparing to TD3, how does the policy performance change if sigma is fixed and it only learns the mean? (mimicing how TD3 learns).
What happens if TD3 is trained like in FB-CPR (Tirinzoni et al). There the agent predicts the mean and the action is sampled through a gaussian distribution. The Q policy gradients can still flow through this sampling (reparam trick). In this comparison, TD3 could predict mean and std.

**Questions:**

When comparing to TD3, how does the policy performance change if sigma is fixed and it only learns the mean? (mimicing how TD3 learns).
What happens if TD3 is trained like in FB-CPR (Tirinzoni et al). There the agent predicts the mean and the action is sampled through a gaussian distribution. The Q policy gradients can still flow through this sampling (reparam trick). In this comparison, TD3 could predict mean and std.

---

> ### Author Response · Authors · 2025-11-21
> **Rebuttal**
>
> We appreciate the reviewer’s insightful comments and thoughtful suggestions. We have carefully examined each point and provide clarifications and revisions to strengthen the paper.
>
> > **Q1 & W2**: When comparing to TD3, how does the policy performance change if sigma is fixed and it only learns the mean? (mimicing how TD3 learns). What happens if TD3 is trained like in FB-CPR (Tirinzoni et al). There the agent predicts the mean and the action is sampled through a gaussian distribution. The Q policy gradients can still flow through this sampling (reparam trick). In this comparison, TD3 could predict mean and std.
>
> **Response**: Thank you for the insightful question. We agree that both variants – (i) learning only the mean with a fixed σ and (ii) predicting both mean and σ with reparameterized sampling – are important comparisons. Both settings are included in our experiments.
>
> 1. **Standard TD3** (CleanRL implementation; Huang et al., 2022) corresponds exactly to the first case: the policy outputs only the mean and uses fixed exploration noise $\sigma$.
>
> 2. **RP-AC** corresponds to the reviewer’s second suggestion. RP-AC is a TD3-style actor–critic where the policy predicts both mean and standard deviation. Similar to TD3, the action is sampled using the reparameterization (RP) trick so that the Q-gradient flows through the sampling step. Details are provided in Appendix D.1 in the updated version (Appendix B.1 in the original submission).
>
> Empirically, RP-AC performs similarly to standard TD3 in our settings, suggesting that the additional stochasticity alone does not close the performance gap with DA-AC.
>
> Huang, S., Dossa, R. F. J., Ye, C., Braga, J., Chakraborty, D., Mehta, K., & AraÃšjo, J. G. (2022). CleanRL: High-quality single-file implementations of deep reinforcement learning algorithms. JMLR.
>
>
> > **W1**: While the authors show the interpolated critic is an important feature and very nicely present how this helps the critic learn a better optimization landscape, it isn't clear what part of the agent is benefitting from this method.
>
> **Response**: Thank you for pointing this out. We clarify that the *policy* is the primary component that benefits from the interpolated critic. The actor updates its distribution parameters by ascending the gradient of $Q_{{w}}(s, u)$, so any improvement in the critic’s curvature directly affects the quality of the policy update. In particular:
>
> 1. **Improved value signal for the actor.** The policy seeks distribution parameters that achieve high value under the critic. A better-behaved critic – one that more accurately reflects how nearby distribution parameters affect returns – provides a more informative and reliable gradient signal for the actor to improve.
>
> 2. **More useful curvature for optimization.** As illustrated in Figure 2, the interpolated critic exhibits significantly more informative curvature compared to the standard critic. This enables faster policy optimization.
>
> We have made this connection between the interpolated critic and the actor’s learning dynamics clearer in the updated version in Section 4.3.
>
> We appreciate the reviewer’s time and feedback. We hope our responses clarify the key points.

---

### Official Review · Reviewer_5jaa · 2025-11-05

**Soundness:** 3
**Presentation:** 4
**Contribution:** 3
**Rating:** 4
**Confidence:** 5

**Summary:**

This work focuses on the policy distributions and proposes a distributions-as-actions framework, an alternative to the classical RL formulation that treats the parameters of parameterized distributions as actions. By shifting this agent-environment boundary, they develop a continuous-action algorithm for a diverse class of action spaces and achieve lower variance. Across experimental results, this paper achieves competitive performance.

**Strengths:**

The motivation is novel and interesting, in which parameterized action distributions are an important topic in the RL community.
The reparameterization policy gradient estimator can reduce variance using interpolated critic learning.
And under selected baselines, this work achieves competitive performance.

**Weaknesses:**

1. The baselines are weak, only SAC, TD3. More baselines should be considered in various settings across discrete, continuous, and hybrid control. The newest baseline was published in 2019.

2. Compared with continuous actions, will discrete action spaces affect performance?

3. The related works on parameterized action distributions are insufficient.

Discretizing continuous action space for on-policy optimization, AAAI, 2020

Discretizing Continuous Action Space With Unimodal Probability Distributions for On-Policy Reinforcement Learning, IEEE TNNLS, 2024.

**Questions:**

See weakness.

---

> ### Author Response · Authors · 2025-11-21
> **Rebuttal**
>
> We thank the reviewer for their comments. We address the concerns point-by-point and provide clarifications below.
>
> > **W1**: The baselines are weak, only SAC, TD3. More baselines should be considered in various settings across discrete, continuous, and hybrid control. The newest baseline was published in 2019.
>
> **Response**: We appreciate the reviewer’s suggestion. SAC and TD3 were chosen because they remain the dominant and widely accepted baselines for continuous-control evaluation in both Gym MuJoCo and DM Control. They continue to serve as default baselines even in recent works (e.g., Fujimoto et al. 2023; D’Oro et al., 2024; Lee et al., 2025; Castanyer et al., 2025).
>
> Importantly, many newer RL developments, including the above works, focus on representation learning, large-scale architectures, or offline/model-based settings, which are largely orthogonal to online RL algorithmic design. Since our goal is specifically to analyze algorithmic differences within online RL, SAC, TD3 and their respective variants provide strong, stable, and standardized baselines that remain the most appropriate for our evaluation settings.
>
> Fujimoto, S., Chang, W. D., Smith, E., Gu, S. S., Precup, D., & Meger, D. (2023). For sale: State-action representation learning for deep reinforcement learning. NeurIPS.
>
> D'Oro, P., Schwarzer, M., Nikishin, E., Bacon, P. L., Bellemare, M. G., & Courville, A. (2024). Sample-Efficient Reinforcement Learning by Breaking the Replay Ratio Barrier. ICLR.
>
> Lee, H., Hwang, D., Kim, D., Kim, H., Tai, J. J., Subramanian, K., ... & Seno, T. (2025). SimBa: Simplicity Bias for Scaling Up Parameters in Deep Reinforcement Learning. ICLR.
>
> Castanyer, R. C., Obando-Ceron, J., Li, L., Bacon, P. L., Berseth, G., Courville, A., & Castro, P. S. (2025). Stable Gradients for Stable Learning at Scale in Deep Reinforcement Learning. NeurIPS.
>
> > **W2**: Compared with continuous actions, will discrete action spaces affect performance?
>
> **Response**: Thank you for the question. Yes, representing the same control problem using discrete versus continuous action spaces can influence performance. In our framework, performance can vary because the distribution families and parameterizations differ across different action spaces, which in turn affects the policy optimization landscape and critic approximation.
>
> To quantify this, we compared DA-AC (Continuous) and DA-AC (Discrete) across Gym MuJoCo and DMC environments, and the per-environment results are reported in Figure 12 of the appendix. We observe that the two variants perform similarly in many cases, but each can outperform the other depending on the environment. When their performance varies, DA-AC (Continuous) more often achieves higher returns, but there are also clear counterexamples (e.g., fish-swim) where DA-AC (Discrete) performs better. This suggests that the choice of action-space representation interacts with environment dynamics and can bias optimization positively or negatively.
>
> We have clarified this point in Appendix A in the updated version. In addition, we have also included a discussion on related work that discretize the action space for continuous control.
>
> > **W3**: The related works on parameterized action distributions are insufficient.
>
> **Response**: We appreciate the reviewer’s comment. We have included an extended related work discussion in Appendix A that covers those on parameterized actions.
>
> We hope that our clarifications and revisions address all concerns raised. We believe the updated results further strengthen the contributions of the paper.

---

### Meta-Review · Area_Chair_Z8mT · 2026-01-04

**Summary:**

This work considers the problem of action representation in reinforcement learning. It proposes to treat the parameters of probability distributions as the actions in an RL setting. This enables the representation of numerous different action spaces in a unified, continuous way due to the continuous nature of the distribution parameters. The work proposes a corresponding policy gradient estimator and an actor-critic algorithm that can handle the continuous action space. The action space itself is implemented by treating the sampling process as part of the environment.

**Reviewer Concerns:**

The reviewers were largely on the positive side. The most prevalent concern centered on positioning and related works (5jaa, WRjw). The reviewers also raised questions about the baselines and comparisons (5jaa, 8RUo, Xv1R).

**Reviewer Scores:**

Many of the reviewers' comments were addressed in the author's response. Probably the one where it is not clear how satisfied the reviewer would be with the response is the comment about the weakness of the baselines.

---

### Decision · Program_Chairs · 2026-01-26

Accept (Poster)